

# Environmental and biological controls on Na/Ca ratios in
# scleractinian cold-water corals
Nicolai Schleinkofer[1], Jacek Raddatz[1], André Freiwald[2,3], David Evans[1], Lydia Beuck[2], Andres
Rüggeberg[4], Volker Liebetrau[5]
[1]Goethe University Frankfurt, Institute of Geosciences, Altenhöferallee 1, 60438 Frankfurt am Main,
Germany
[2]Senckenberg am Meer, Marine Research Department, Südstrand 40, 26382 Wilhelmshaven, Germany
[3]5MARUM (Zentrum für Marine Umweltwissenschaften), Bremen University, Leobener Str. 8, 28359
Bremen, Germany
[4]Department of Geosciences, Faculty of Science and Medicine, University of Fribourg, Chemin du
Musée 6, CH-1700 Fribourg, Switzerland
[5]GEOMAR Helmholtz Centre for Ocean Research Kiel, Wischhofstr. 1-3, D-24148 Kiel, Germany
Correspondence to: Nicolai Schleinkofer (schleinkofer@em.uni-frankfurt.de)





**Abstract**
Here we present a comprehensive attempt to correlate aragonitic Na/Ca ratios from *Lophelia*
*pertusa, Madrepora oculata* and a caryophylliid cold-water coral (CWC) specie*s* with different
seawater parameters such as temperature, salinity and pH. Living CWC specimens were
collected from 16 different locations and analyzed for their Na/Ca content using solution-based
inductively coupled plasma-optical emission spectrometry (ICP-OES) measurements.
The results reveal no apparent correlation with salinity (30.1–40.57 g/kg) but a significant
inverse correlation with temperature (-0.31 mmol/mol/°C). Other marine aragonitic organisms
such as *Mytilus edulis* (inner aragonitic shell portion) and *Porites* sp. exhibit similar results
highlighting the consistency of the calculated CWC regressions. Corresponding Na/Mg ratios
show a similar temperature sensitivity to Na/Ca ratios, but the combination of two ratios appear
to reduce the impact of vital effects and domain-dependent geochemical variation. The high
degree of scatter and elemental heterogeneities between the different skeletal features in both
Na/Ca and Na/Mg however limit the use of these ratios as a proxy and/or make a high number
of samples necessary. Additionally, we explore two models to explain the observed
temperature sensitivity of Na/Ca ratios for an open and semi-enclosed calcifying space based
on temperature sensitive Na and Ca pumping enzymes and transport proteins that change the
composition of the calcifying fluid and consequently the skeletal Na/Ca ratio.
**1. Introduction**
Sodium-calcium ratios (Na/Ca) are a promising new tool in palaeoceanography to reconstruct
seawater salinities. Cultured benthic and planktonic foraminifera as well as living planktonic
foraminifera from the Red Sea showed the potential of calcitic Na/Ca ratios as an independent
salinity proxy (Mezger et al., 2016; Wit et al., 2013). Cold-water corals provide one of the most
promising marine paleoenvironmental archives for climatic research due to the potential to
reconstruct high-resolution records using the aragonitic skeleton. About half of the known
scleractinian coral species do not live in tropical, shallow water (<50 m) but in deeper waters,
including deep-sea environments (>200 m) (Roberts et al., 2009). These deeper or cold-water



corals lack phototrophic symbionts and therefore are azooxanthellate. Like their zooxanthellate
shallow-water relatives, some azooxanthellate deeper water species, such as *Lophelia pertusa*
and *Madrepora oculata,* are also capable of building large three-dimensional reef frameworks
that serve as habitats for thousands of different organisms and constitute biodiversity hotspots
in low to high latitudes and from shallower water to the deep seas (Henry and Roberts, 2016;
Roberts et al., 2009). In contrast to shallow-water corals, cold-water corals are not bound to
the photic zone. Instead their distribution is controlled by several parameters, amongst which
is the density of seawater (mainly controlled by salinity)(Dullo et al., 2008) as it seems to
correlate with the so called Intermediate Nepheloid Layers (INL) which contribute an important
source of particulate organic matter (POM) (Kiriakoulakis et al., 2005, 2007). Additionally, it
has been suggested, that gamete density restricts the lateral transport to certain density
envelopes (Dullo et al., 2008). For *L. pertusa*, the suitable density envelope amounts to $\sigma_\theta =$
27.35–27.65 kg/m$^3$ (Dullo et al., 2008), although these values are not applicable to every
oceanic region (Flögel et al., 2014; Rüggeberg et al., 2011). Since seawater density is a
function of temperature and salinity, these parameters also partly control the spatial
distribution. Most known CWC reefs occur in salinities of 35 g/kg and mean temperatures of
4–12°C (Freiwald, 2002; Freiwald and Roberts, 2005), but they are also able to thrive in lower
and higher temperatures and salinities (e.g. Bett, 2001; Roder et al., 2013; Taviani et al., 2005).
Oxygen saturation does not seem to have a strong effect on occurrences, as CWC survive in
oxygen concentrations from at least 2.6 to 6.7 ml/l (Schroeder, 2002; Wisshak et al., 2005) and
even tolerate short periods in hypoxic conditions (Dodds et al., 2007). While the aragonite
saturation, the product of [Ca$^{2+}$] and [CO$_3$$^{2-}$] divided by the solubility product of aragonite, is of
great importance and should be generally high, CWC possess mechanisms to raise their
internal aragonite saturation and pH (McCulloch et al., 2012; Raddatz et al., 2014b; Rollion-
Bard et al., 2011) through ion pumps (Kingsley and Watabe, 1985). The resistance of CWC to
changes in the carbonate system is therefore rather high (McCulloch et al., 2012). To facilitate
the internal regulation, a constant nutrient supply is necessary to fulfill the energy requirements
(McCulloch et al., 2012). Most CWC reefs are located in areas with low nutrient concentrations



(Davies et al., 2008) such that they mostly rely on POM as their main nutrient supply
(Kiriakoulakis et al., 2005, 2007) and therefore need strong currents that provide a constant
supply of organic matter (Kiriakoulakis et al., 2007; Mortensen et al., 2001). The reliance on
POM might also be a reason for the occurrence of CWC in certain density envelopes because
of their correlation with INL's providing POM (White et al., 2005).
Independent proxies are needed to reconstruct living conditions of CWC in the past to better
understand their temperature/salinity/pH tolerances and to research the influence of these
conditions on the spatial distribution. This would help to better locate new unknown sites of
CWC occurrences. For temperature and pH, different geochemical proxies can be used to
calculate these parameters in the geological past. Sr/Ca and Mg/Li ratios serve as temperature
proxies (Cohen et al., 2006; Gagnon et al., 2007; Mitsuguchi et al., 1996; Montagna et al.,
2014; Raddatz et al., 2013, 2014a; Rollion-Bard and Blamart, 2015; Shirai et al., 2005), U/Ca
and Boron-isotopes serve as proxies of the carbonate system(Anagnostou et al., 2011, 2012;
Blamart et al., 2007; McCulloch et al., 2012; Raddatz et al., 2014b, 2016; Rollion-Bard et al.,
2011). Independent geochemical methods to reconstruct past salinities however are absent
but urgently needed to reconstruct spatial distribution patterns in the past and quantify effects
of ocean acidification on CWC. Even though CWC show that they can maintain growth in
under-saturated, corrosive waters, the older unprotected parts of the reef are susceptible for
dissolution (Büscher et al., 2017; Form and Riebesell, 2012).This weakens the reef integrity
and might cause severe implications on available microhabitats (Büscher et al., 2017; Roberts,

91  2006)

Reconstructing past salinities can be accomplished with several different techniques, e.g.
diatom and dinoflagellate species composition(Zonneveld et al., 2001), morphology and size
of placoliths from *Emiliana huxleyi* (Bollmann et al., 2009)*,* Ba/Ca ratios in foraminiferal calcite
(Weldeab et al., 2007), strontium isotope composition in bivalves (Israelson and Buchardt,
1999), process length of dinoflagellate cysts (Mertens et al., 2009), hydrogen isotope
composition of alkenones (van der Meer et al., 2007; Schouten et al., 2006) or temperature



corrected (Mg/Ca, TEX86) oxygen isotopes (Elderfield and Ganssen, 2000). While some of
these proxies may yield reliable results (e.g. coupled Mg/Ca and oxygen isotopes (Elderfield
et al., 2012; Lear et al., 2000)) others suffer from rather large uncertainties introduced by
modelled parameters or require a good knowledge of the regional oceanography (Wit et al.,
2013). Others, like Ba/Ca ratios are more effected by terrestrial runoff and are therefore only
applicable in proximal sites. Complications with the existing proxies mean that further methods
are desirable, here we explore whether coral Na/Ca ratios may be useful in this regard.
The influence of seawater salinity on Na/Ca ratios are known from Atlantic oysters (Rucker
and Valentine, 1961), barnacle shells (Gordon et al., 1970) as well as inorganically precipitated
calcium carbonate (Ishikawa and Ichikuni, 1984; White, 1977). Recently it has been shown
that Na incorporation in calcitic planktonic and benthic foraminifera appears to be largely
controlled by seawater salinity (Allen et al., 2016 (only in *Globigerinoides ruber*); Mezger et al.,
2016; Wit et al., 2013). According to Wit et al. (2013), the incorporation of Na in calcite is
dependent on the activity of Na in the seawater which is a function of the salinity. There is
strong evidence that Na does substitute for Ca in biogenic aragonite despite its charge
difference (Okumura and Kitano, 1986; Yoshimura et al., 2017). Since Na and Ca compete for
the same lattice positions, the calcium concentration and Na/Ca ratio of the surrounding
seawater might control the amount of sodium incorporation (Ishikawa and Ichikuni, 1984;
White, 1977). This would inhibit the use of Na/Ca ratios as a salinity proxy but might prove
useful to reconstruct oceanic calcium concentrations. Recent studies also show that the Na/Ca
ratio in foraminiferal calcite is also mainly controlled by seawater Na/Ca ratios (Hauzer et al.,

119   2018).

In this study, we investigate the impact of different seawater parameters on the incorporation
of Na in the aragonitic skeleton of the scleractinian cold-water coral *L. pertusa, M. oculata* and
a caryophylliid species from the Red Sea. The corals were collected alive from a variety of
locations to cover a broad range of temperatures (5.9–21.6°C) and salinities (30.1–40.6 g/kg).
**2. Materials & Methods**



### 2.1. Study area and sample collection


The samples were taken from **45** different coral specimens that were collected from 16 different
locations (Tab. 1). Most of the samples (**n=25**) were collected during different cruises from the
Norwegian margin. The other samples derive from the Irish Margin and Bay of Biscay (**n=4**),
the Mediterranean Sea and Gulf of Cadiz (**n=7**), the Gulf of Mexico and Great Bahama Bank
(**n=4**) and the Red Sea (**n=5**) (Fig. 1). Conductivity-Temperature-Depth (CTD) downcast data
for water parameters was available for all locations except the Red Sea and the Gulf of Mexico.
Where no CTD data was available, the water parameters were retrieved from annual averaged
data from World Ocean Atlas 2013. Where available, comparison of in-situ CTD and WOA13
data, revealed an agreement within 0.15°C in Santa Maria de Leuca and 0.04°C in the Bay of
Biscay respectively. The seawater carbonate system data such as pH was taken from the
associated cruise report (Flögel et al., 2014).
We took 31 samples from different coral colonies and three different species (*L. pertusa, M.*
*oculata, Caryophyllia* sp.) that were collected during different cruises. The samples were taken
from the uppermost calices after physically cleaning them with a dental drill tool to remove
secondary overgrowths. We avoided further cleaning or rinsing with water because studies
suggest that structurally substituted Na is readily leached even by distilled water (Ragland et
al., 1979). It is possible that organic contents inside the skeleton bias the results (Branson et
al., 2016). However, the study on foraminifera shows that the Na/Ca ratio only significantly
varies at POS (primary organic sheet) regions. In corals the COC (centers of calcification)
would be an equivalent structure, which was tried to avoid during the sampling process.
Furthermore, it is stated that these regions only significantly affect bulk sample elemental ratios
in very thin walled foraminifera (Branson et al., 2016). In corals the area of COC is rather large
(20% of the total skeleton radius (Rollion-Bard and Blamart, 2015)) but the Na/Ca ratio does
not increase in the COC as strong as it does in the POS areas of foraminifera (Branson et al.,
2016; Rollion-Bard and Blamart, 2015). Avoiding the COC areas in bulk samples only reduces
the mean Na/Ca ratio by 0.18 mmol/mol, additional cleaning of organic material is therefore



not necessary. An additional 14 samples were prepared as longitudinal slices through the
corals calice, glued on metal plates. In order to identify elemental heterogenities within the
corals theca wall, subsamples were taken using the micromill (Merchantec MM-000-134).
**2.2. ICP-OES Analyses**
The elemental ratios were measured with inductively coupled plasma optical emission
spectrometry (ICP-OES). The ICP-OES analysis was carried out with a Thermoscientific iCap
6300 dual viewing at Goethe University/Frankfurt. This machine is both capable of measuring
axially and radially. Alkali metals (Na) were measured radially on line 589.59 nm whereas
earth-alkali metals (Mg, Sr) were measured axially on lines 279.55 nm and 421.55 nm
respectively. The sample powder (≈ 140 µg) was dissolved in 500 µl $HNO_3$ (2%) and 300 µl
aliquots were separated. Subsequently 1500 µl of 1.2 mg/l Yttrium solution was added to each
aliquot as an internal standard resulting in 1 mg/l. The intensity data was background
subtracted and standardized internally to Y and normalized to Ca. External standards were
mixed from single element standard solutions to match the typical element concentrations of
cold-water corals (cf. Rosenthal et al., 1988). The coral standard JCp-1 (Hathorne et al., 2013;
Okai et al., 2002) was measured after every tenth sample to allow for drift correction and
monitor measurement quality.
Relative precision of the Element/Ca measurements was based on the international calcium-
carbonate standard JCp-1 (20 replicates) and amounts to 20.47 ± 0.68 mmol/mol Na/Ca (19.8
± 0.14 mmol/mol (Okai et al., 2002)), 4.09 ± 0.11 mmol/mol Mg/Ca (4.199 ± 0.065 mmol/mol
(Hathorne et al., 2013; Okai et al., 2002)) and 9.36 ± 0.07 mmol/mol Sr/Ca (8.838 ± 0.042
mmol/mol (Hathorne et al., 2013; Okai et al., 2002)). Following from this relative precision is
better than 4 %. Accuracy amounts to 103 ± 3% Na/Ca, 97 ± 3% Mg/Ca and 106 ± 0.7 %
Sr/Ca. Measurements were conducted in two sessions lasting ten and five hours.
**2.3. Data Processing**



Before calculations of correlations or applying statistics outliers were removed from the raw
data. Outliers were identified by the average ±1.5 SD per oceanic region (Norwegian margin,
Bay of Biscay/Irish Margin, Mediterranean Sea, Red Sea, Gulf of Mexico/Bahamas). The
threshold was chosen to cover a range from 15 to 35 mmol/mol which is roughly 5 mmol/mol
higher and lower than the reported range from a similar study (Rollion-Bard and Blamart,
2015). The profiled samples were additionally checked for values that derive from the COC,
which are identifiable through a positively correlating increase in Mg/Ca and Na/Ca. The
chosen threshold was the mean of the profiled sample + 2SD of the JCp-1. These values are
not used for further calculations as well. Statistical calculations were conducted with the
ORIGIN Pro software suite.

**3. Results**

Spatial distribution patterns show great variations in Na/Ca ratios through the corals skeleton
(Fig. 2). In the COC and COC-like structures Na/Ca ratios show significant increases but the
amount of increase relative to the mean is not uniform in the sample. Increases range from +2
to +10 mmol/mol. Mg/Ca is positively correlated with Na/Ca in the COC structures but mostly
independent from each other in the fibrous deposits (FD). Similar to Na/Ca, the amplitude of
Mg/Ca in the COC-structures is very variable in their amount and ranges from +0.5 to +3
mmol/mol. Both sodium and magnesium are often enriched in the outermost parts of the theca.
Sr/Ca ratios are mostly stable throughout the theca and seem to be independent from the
different skeletal structures. In some samples, co-variances are present but in general they do
not appear to be controlled by the skeletal morphology in the same way as Mg/Ca and Na/Ca
as shown by their independency from the different skeletal structures.

**3.1 Element/Ca ratios of scleractinian cold-water corals**

Na/Ca ratios vary between 20.49 mmol/mol in the Red Sea and 31.04 mmol/mol in the
Norwegian reefs with a mean at 25.22 mmol/mol and a standard deviation of 2.8 mmol/mol
(Fig. 3). The values are in accordance to previous studies on *L. pertusa* (21.94–28.11
mmol/mol (Rollion-Bard and Blamart, 2015)), but 5 mmol/mol higher than reported for





zooxanthellate corals (Amiel et al., 1973; Busenberg and Niel Plummer, 1985; Mitsuguchi et
al., 2001; Ramos et al., 2004; Swart, 1981). Significant deviations between *L. pertusa (n=38)*,
*M. oculata* (n=2) and *Caryophyllia* sp. (n=5) are not observable. A linear correlation between
salinity and Na/Ca over the whole salinity range is not observable, but the present dataset is
best described with a second order polynomial function. Accordingly, there is a positive trend
from 30.1–35 g/kg followed by a negative trend from 35–40.5 g/kg. Linear regressions equal:
$f(S_{30.1-35}) = 6.4 + 0.56 * S$ ($R^2 = 0.99$, *P* = 0.072) and $f(S_{35-40.5}) = 56.61 − 0.84 * S$ ($R^2 = 0.66$,
*P*= 0.4). As the *P*-values show a significant slope is missing in all these regressions. In the
case of the polynomial fit the *P*-value shows that the fit is not significantly superior to $f(S_{30,1-40,5})$
$_{40,5})$ = constant.
Na/Ca and temperature show a significant negative correlation. The linear regression equals:
$$f_{T\ 6-22°C} = 28.2 \pm 0.9 - 0.31 \pm 0.07 \times T \ (R^2 = 0.87, P = 0.02) \tag{1}$$
Temperature and salinity show a positive correlation so the correlation between Na/Ca and
temperature is not caused by a negative correlation between salinity and temperature. Corals
from the Mediterranean Sea are slightly elevated in their Na/Ca ratio but within the error they
still fit the correlation with both salinity and temperature. Distribution coefficients ($K_d^{Na}$ =
Na/Ca$_{carbonate}$/ Na/Ca$_{seawater}$) at specific temperatures for several different species, including the
scleractinian cold-water corals from this study, *Porites* sp. and *M. edulis*, show similar values.
$K_d^{Na}$ from this study amounts to $K_d^{Na}{}_{(6.23°C)} = 5.73*10^{-4}$, $K_d^{Na}{}_{(7.94°C)} = 5.51*10^{-4}$, $K_d^{Na}{}_{(9.83°C)}$ =
$5.44*10^{-4}$, $K_d^{Na}{}_{(13.54°C)} = 5.62*10^{-4}$, $K_d^{Na}{}_{(21.64°C)} = 4.73*10^{-4}$. Distribution coefficients for *Porites*
sp. and *M. edulis* are $K_d^{Na}{}_{(26.03°C)} = 4.6*10^{-4}$ (Mitsuguchi et al., 2001) and $K_d^{Na}{}_{(12.5°C)} = 5.25*10^{-4}$
$^{-4}$ (Lorens and Bender, 1980) respectively. Inorganic distribution coefficients are with $4.00*10^{-4}$
about 20% lower in comparison (Kinsman, 1970).The results from White (1977) show that the
composition of the solution affects the elemental ratios in the precipitate, but in the study from
Kinsman (1970) the precipitation happened from seawater. Therefore, it is reasonable to use
this data for comparison. A combined regression using the data from this study, the *L. pertusa*



data from Rollion-Bard and Blamart (2015), *M. edulis* data from Lorens and Bender (1980) and
*Porites* sp. data from Ramos et al. (2004) and Mitsuguchi et al. (2001) equals:
$f_{T\,6-27.63\,°C} = 28.03 \pm 0.7 - 0.31 \pm 0.04 \times T\ (R^2 = 0.9, P < 0.0001)$         (2).
Values for Na/Ca also show a significant positive correlation with pH of the ambient seawater.
Linear regression equals: $f(pH) = -84.26 + 13.63 * pH$ ($R^2 = 0.14$, $P = 0.017$). A correlation
between pH and temperature is absent.
**3.2. Mg/Ca & Sr/Ca**
Mg/Ca values vary between 2.2 mmol/mol in the Red Sea and 6.38 mmol/mol in the
Mediterranean Sea with a mean of 3.99 mmol/mol and a standard deviation of 0.97 mmol/mol
(Fig. 4). Maximum values are higher than literature states for *L. pertusa* (2.99–4.72 mmol/mol
(Raddatz et al., 2013)) but the mean values are well inside the range of literature. Significant
deviations between *L. pertusa*, *M. oculata* and *Caryophyllia* sp. are not observable. Seawater
parameters such as water temperature, salinity and pH have no significant effect on Mg/Ca
ratios in the skeleton.
Sr/Ca values vary between 9.46 and 10.46 mmol/mol with a mean of 10.1 mmol/mol and a
standard deviation of 0.25 mmol/mol (Fig. 5). Both maximum and minimum values derive from
corals that grew in reefs that are located in the Trondheimfjord. The values are in accordance
to previous studies on *L. pertusa* (9.27–10.05 mmol /mol (Raddatz et al., 2013)). Significant
deviations between *L. pertusa*, *M. oculata* and *Caryophyllia* sp. are not observable. Despite
the known temperature effect on Sr/Ca ratios this effect is not pronounced in this dataset. The
correlation shows a strongly deviating slope of -0.015 mmol/mol/°C in comparison to -0.083 ±
0.017 mmol/mol/°C, which is given in literature(Raddatz et al., 2013). Linear regressions equal:
$f(T) = 10.26 - 0.015 * T$ ($R^2 = 0.83$, $P = 0.03$)., Sr/Ca vs. salinity values show a similar
distribution pattern like Na/Ca vs. salinity values with the maximum at 35 g/kg and descending
values at lower and higher salinities but an AIC and a F-Test confirm that a linear fit is better



suited. The Linear regression equals f(S) = 10.58 – 0.015 * S (R² = 0.52, *P* = 0.17). *P*-values
show that the correlation is not significant.

**3.3 Elementconcentration in the extracellular calcifying fluid (ECF)**

Based on the assumption of a semi-enclosed ECF with seawater-leakage and a consequent
$[Na]_{ECF}$ similar to $[Na]_{Seawater}$ it is possible to calculate $[Ca]_{ECF}$ and $[Mg]_{ECF}$ using skeletal Na/Ca
and Mg/Ca data. Assuming $[Na]_{Seawater}$ = $[Na]_{ECF}$ = 455 mmol/l (Turekian et al., 2010) and an
invariant Na distribution coefficient, $[Ca]_{ECF}$ can be calculated with the following equation:
$$[Ca]_{ECF} = [Na]_{ECF} * \frac{K_d^{Na}}{\frac{Na}{Ca}_{Coral}}$$    (3)
In order to do so, knowledge of $K_d^{Na}$ is required. White (1977) reports 1.8 – 4.1*10$^{-4}$ for inorganic
aragonite in the four experiments with solution Na/Ca closest to the natural seawater ratio (~45
mol/mol), which would result in predicted aragonite Na/Ca ratios of 8 – 18 mmol/mol, slightly
lower than the coral aragonite values we measure. Because this difference may be explained
via differences in (e.g.) inorganic and coral aragonite growth rates or the presence of organics,
we adjust our data so that the mean [Ca] value lies close to seawater (~10 mmol/l) by using
$K_d^{Na}$=5.37*10$^{-4}$ calculated from the coral samples presented here. As such we cannot presently
constrain absolute $[Ca]_{ECF}$ values using this method, however the aim here is simply to explore
whether differences in $[Ca]_{ECF}$ can explain the variance in both our Na/Ca and Mg/Ca data. An
improved understanding of the inorganic distribution coefficient may enable both precise and
accurate ECF reconstructions in the future. Using the method outlined above, we calculate
$[Ca]_{ECF}$ values ranging from 7.9 mmol/l to 12.3 mmol/l with a mean of 9.9 mmol/l. This range is
in good agreement with the microsensor studies on *Galaxea fascicularis* conducted by Al-
Horani et al., (2003)(9-11 mmol/l). By substituting these data into the equation:
$$[Mg]_{ECF} = \frac{Mg}{Ca}_{Coral} * \frac{[Ca]_{ECF}}{K_d^{Mg}}$$    (4)
With $K_d^{Mg}$=7.9*10$^{-4}$, calculated from the coral samples presented here, $[Mg]_{ECF}$ can also be
calculated. Resulting values range from 32.8 mmol/l to 104.7 mmol/l and a mean of 51.5 mmol/l




and a median of 46.5 mmol/l. Results show that the Mg-concentration in the ECF is constant
with changing Ca-concentration.
**4. Discussion**
**4.1 Heterogeneities of elemental ratios in scleractinian corals**
Ninety percent of the sodium in corals is located in the aragonitic mineral phase, the remaining
sodium is bound to organic material and exchangeable sites (Amiel et al., 1973). Magnesium,
which co-varies with sodium, is not located in the aragonitic phase but either organic material
(20–30%) and a highly disordered inorganic phase such as amorphous calcium carbonate
(ACC) (70–80%) (Amiel et al., 1973; Finch and Allison, 2008) or nanodomains of Mg-bearing
carbonate occluded in the aragonite (Finch and Allison, 2008). A small percentage seems to
be also trapped along the (001) surface (Ruiz-Hernandez et al., 2012). Elemental
heterogeneities are particularly visible when comparing COC and fibrous deposits (Fig. 2).
COC are both chemically and morphologically distinct from the fibrous deposits. While the
COC's are built by sub-micron sized granular crystals(Constantz, 1989), the fibres that build
the fibrous zones are not single orthorhombic crystals but elongated composite structures with
very fine organo-mineral alternations (Cuif and Dauphin, 1998). Reasons for the different
chemical composition are still under debate and include: (1) pH variations in the calcifying fluid
(Adkins et al., 2003; Holcomb et al., 2009), (2) Rayleigh fractionation (Cohen et al., 2006;
Gagnon et al., 2007), (3) kinetic fractionation (McConnaughey, 1989; Sinclair et al., 2006), (4)
mixed ion transport through direct seawater transport and ionic pumping (Gagnon et al., 2012),
and (5) precipitation from different compartments (Meibom et al., 2004; Rollion-Bard et al.,

301 2010, 2011).

The missing co-variance between Sr/Ca and Mg/Ca or Na/Ca ratios excludes Rayleigh
fractionation as the main mechanism responsible for the large variances of elemental ratios
(Rollion-Bard and Blamart, 2015), as well as mixed ion transport for similar reasons (Rollion-
Bard and Blamart, 2015). pH variations and consequent changes in the saturation of the
calcifying fluid have been shown to alter Mg/Ca ratios in corals and abiogenic aragonite



(Holcomb et al., 2009) and therefore, could potentially alter Na/Ca ratios as well. While the pH-
elevation at the COC is supported by several studies (Al-Horani et al., 2003; Raddatz et al.,
2013; Rollion-Bard et al., 2011), Tambutté et al. (2007) propose that the nanometer sized
spaces between the skeleton and the calicoblastic ectoderm does not allow a modification of
the saturation state. Our data may be explained by different calcification compartments in
combination with kinetic effects caused by rapid calcification rates. Additionally, we propose
changing organic contents as a further mechanism that controls elemental ratio differences in
the different skeletal parts, visible in the covariance of Mg/Ca and Na/Ca ratios throughout the
skeleton. It is not clear in which way the different precipitation regions discern from each other,
different cell types or different modes of the same cell types (Rollion-Bard et al., 2010). So far,
only calicoblasts and desmocytes are known from the aboral ectoderm of corals (Allemand et
al., 2011; Tambutté et al., 2007) but calicoblasts show differences in their morphology, ranging
from very thin, long and flat to thick and cup like (Tambutté et al., 2007). A major controlling
factor on the cell shape is the calcification activity, with flat calicoblasts corresponding to low
calcification activity and thick calicoblasts to high calcification activity (Tambutté et al., 2007).
These different cell morphologies might be the reason for different types of precipitation, ACC,
a proposed precursor phase of aragonite (Von Euw et al., 2017; Rollion-Bard et al., 2010), and
granular crystals in the COC regions or organo-mineral fibres in the fibrous deposits. The
precipitation of ACC in the COC would certainly explain the enrichment of Mg in these areas,
as it is necessary to stabilize the otherwise unstable ACC (Von Euw et al., 2017). Furthermore,
the COC's are known to be rich in organic material (Cuif et al., 2003; Stolarski, 2003), also
explaining the enrichment of Mg as well as explaining a slight enrichment of Na. However, the
amount of Na bound to organic material is not high enough (Amiel et al., 1973) that the
enrichment in the COC can be solely explained by high organic contents. Kinetic effects, due
to rapid calcification rates are more likely to be the main control for Na variations in COC and
fibrous deposits. Since Na is incorporated in the aragonite lattice by direct substitution with Ca
(Okumura and Kitano, 1986; Yoshimura et al., 2017), charge differences occur due to the
exchange of divalent Ca with monovalent Na. These charge differences need to be





compensated by lattice defects/$CO_3^{2-}$ vacancies, which occur more often at higher precipitation
rates (Mucci, 1988; White, 1977; Yoshimura et al., 2017). Growth rate effects are also known
for the incorporation of Mg, albeit these effects are more likely caused by crystal surface
entrapment of Mg by new formed aragonite (Gabitov et al., 2008, 2011; Watson, 1996).
Sr/Ca ratios in the warm-water coral *Pocillopora damicornis* seems to be largely unaffected by
growth rate changes over a range of one to over 50 µm/day (Brahmi et al., 2012), at least when
comparing different skeletal architectures (Fig. 2). This is supported by our data as the
observed Sr/Ca ratios show no significant decrease in the COC or COC-like areas as it would
be excepted from the results of de Villiers et al. (1994) despite the significantly different growth
rates in these areas (COC > 50–60 µm/day, FD = 1–3 µm/day (Brahmi et al., 2012)). In fact,
an increase in the COC is more often but still not regularly, visible (Cohen et al., 2006).
Consequently, a significant effect of the different skeletal architectures on Sr/Ca ratios in
coralline aragonite can be excluded. Slight increases in the COC however can be explained
with the great adsorbtion potential of Sr to organic matter (Chen, 1997; Khani et al., 2012;
Kunioka et al., 2006)

**4.2. Environmental control on coral Na/Ca ratios**

**4.2.1. Salinity**

Recently, Na/Ca ratios in foraminiferal calcite have shown the potential to provide an
independent salinity proxy (Allen et al., 2016; Bertlich et al., 2018; Mezger et al., 2016; Wit et
al., 2013). Na/Ca ratios in foraminiferal calcite show significant positive correlations with the
salinity albeit with species-specific offsets and slopes. Ishikawa and Ichikuni (1984) proposed
that the activity of Na in seawater is the primary controlling factor for the incorporation of Na in
calcite. However, more recent studies have shown that Na/Ca in foraminiferal calcite is mainly
driven by the seawater Na/Ca ratio instead of the Na activity when this is the dominant variable
(Evans et al., 2018; Hauzer et al., 2018). Species-specific offsets make further biological
controls highly plausible.



In this study, no correlation between salinity and Na/Ca ratios is present (Fig. 3). The positive
trend up to 35 g/kg followed by a negative trend after 35 g/kg can be explained by growth rate
changes due to the changing salinity. To our knowledge no studies on the effect of salinity on
growth rates have been conducted on *L. pertusa* but it is plausible that it shows reduced growth
rates in salinities diverging from the biological optimum similar to other marine organisms (e.g.
*M. edulis* (Malone and Dodd, 1967)). A specific osmoregulation is probably not needed for
CWC in the mostly salinity stable habitats they live in (Roberts et al., 2009). Reduced growth
rates consequently lower the amount of lattice defects and the amount of possible
incorporation sites for sodium (Mucci, 1988; White, 1977; Yoshimura et al., 2017).

If Na/Ca ratios in corals are controlled by calcification rates, a calcification rate proxy could be
used to correct this effect. Sr/Ca ratios have been discussed as a possible growth rate proxy
(de Villiers et al., 1994) and may be used to determine changes in growth rate. However, our
data shows that the Sr/Ca ratios remain constant with changing salinities. Accordingly,
concluding from the results of de Villiers et al. (1994) the calcification rate would remain
constant over the whole salinity range. It should be noted that higher growth rates do not
necessarily imply higher calcification rates or vice versa. Higher growth rate can also be
caused by higher organic deposits in the skeleton (Stolarski, 2003). Therefore, a change in
calcification cannot necessary be inferred from changing Sr/Ca ratios. Still, the effects that
growth or calcification rate changes and the different skeletal architectures have on Sr/Ca
ratios in corals is still discussed. There is evidence for positive and negative correlation of
Sr/Ca with growth and calcification rate as well as the different skeletal architectures (Allison
and Finch, 2004; Cohen et al., 2006; Kunioka et al., 2006; Raddatz et al., 2013). It still remains
unknown why there is no persistent Sr/Ca variation between the differential skeletal
architectures (COC, fibrous deposits) in this study despite being visible in several other studies
(Cohen et al., 2006; Gagnon et al., 2007; Raddatz et al., 2013). An explanation could be the
low sampling resolution in the profiled samples and possible mixing of COC and fibrous zone
material. Further research is needed to evaluate the effects of growth and calcification rates
on Sr/Ca ratios in biogenic carbonates.



### 4.2.2. Temperature

A temperature control on Na/Ca ratios has been shown in inorganic precipitated aragonite (White, 1977) and in the planktonic foraminifera *G. ruber* and *G. sacculifer* (Mezger et al., 2016), although temperature and salinity covary in that study. Furthermore, Rollion-Bard and Blamart (2015) suggest a possible temperature control on Na/Ca ratios in the CWC *L. pertusa* and the warm-water coral *Porites* sp. However, the temperature sensitivity in inorganic precipitated aragonite is far lower compared to the biogenic aragonite from CWC including a systematic offset of $K_d^{Na}{}_{(15°C)}= 1.17*10^{-4}$. Interestingly, other marine carbonates (*Porites* sp., *M. edulis*) also fit in the calculated temperature sensitivity. This holds true for biogenic aragonite and biogenic calcite, where *M. edulis* fits into the temperature sensitivity found by Mezger et al. (2016). A combined regression using the data from Evans et al. (2018), Mezger et al. (2016) and Lorens and Bender (1980) reveals a temperature sensitivity of ±0.37 mmol/mol/°C which is strikingly similar to the sensitivity in aragonite of ±0.31 mmol/mol/°C (Fig. 6). The samples that Mezger et al. (2016) used in their study derive from the Red Sea, where a negative correlation between the seawater salinity and seawater temperature exists. They conclude that the salinity effect on Na/Ca ratios and the covariance between salinity and temperature cause the temperature sensitivity of Na/Ca ratios. However, it is also possible that the salinity sensitivity is caused by a temperature effect.

The apparent offset between inorganically precipitated aragonite and biogenic carbonates further implies a biological control on Na incorporation. In contrast to other elements such as Lithium (Montagna et al., 2014), the high correlation between *L. pertusa, M. oculata*, *Caryophyllia* sp. *Porites* sp. and *M. edulis* implies that the Na/Ca variance introduced by these possibly occurring vital effects appear to be similar for all these species. We suggest that similar Na pathways into the calcifying space exist in foraminifera, mussels and scleractinian warm-water as well as cold-water corals and temperature exerts a strong control on the activity of these pathways, altering the sodium availability during calcification. Further controls are

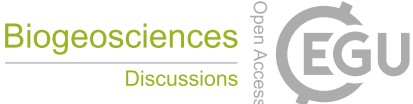

possibly contributed by temperature dependent solubility variations of $CaCO_3$ and $Na_2CO_3$ and
an exothermic Na incorporation mechanism.
Bertlich et al. (2018) proposed that lower temperatures increase the solubility of calcium
carbonate and increase the amount of free Ca, leading to higher Na/Ca ratios at lower
temperatures. Yet such a solubility controlled temperature effect on calcite and aragonite is
rather small, whereas the sensitivity to pressure changes is much more pronounced
(Pytkowicz and Conners, 1964; Zeebe and Wolf-Gladrow, 2001). Accordingly, the Na/Ca ratio
should also decrease with water depth. Here we do observe a relationship between Na/Ca
ratios and water depth, but at constant temperatures (7.2°C – 7.8°C) there is no effect of water
depth (160 m – 280 m) on Na/Ca ratios. The relationship between depth and Na/Ca ratios is
therefore presumably caused by the positive correlation between water temperature and water
depth. A decrease in Na/Ca ratios with temperatures could also be explained by solubility
effects similar to the effects that are discussed to cause the temperature effects on Li/Ca ratios
(Marriott et al., 2004). The solubility of $Na_2CO_3$ increases with increasing temperature (Haynes
et al., 2016). Again, this would result in decreasing Na/Ca ratios with increasing temperature,
because the solubility of $Na_2CO_3$ decreases relative to calcium carbonate (Haynes et al.,
2016), making it thermodynamically less favorable to incorporate Na. The effects of pressure
on the solubility of $Na_2CO_3$ cannot be quantified at the moment due to the lack of studies.
Moreover, the temperature effect can also be caused by an exothermic substitution
mechanism of Na into the aragonite lattice, similar to the incorporation of Mg in calcite (Mucci
and Morse, 1990). If the substitution between Ca and Na is exothermic, consequently the
incorporation of Na is favored at lower temperatures. However, there is to our knowledge, no
study available that contains enthalpy data for this reaction. While the proposed mechanism
by Bertlich et al., (2018) can be excluded as an explanation for the temperature sensitivity of
Na/Ca ratios, the other explanations are equally plausible in terms of the existing studies. Still,
the differences in the temperature sensitivity between inorganic precipitated aragonite and
biogenic aragonite requires further biological controls to explain this deviation.



As an alternative, we explore whether temperature dependent Na membrane pathways can explain temperature effects on aragonitic Na/Ca ratios. There are several enzymes and ion pumps known that constitute sodium pathways through the membrane of the calcifying space. $Na^+/K^+$-ATPase are known from the tropical coral *Galaxea fascicularis* (Ip and Lim, 1991), Na/Ca ion pumps are suggested to exist in *Galaxea fascicularis* and *Tubastraea faulkneri* (Marshall, 1996). Na/K ATPase was found in the bivalve species *M. edulis* and *Limecola balthica* (Pagliarani et al., 2006; Wang and Fisher, 1999) as well as Na/Mg ion pumps in *Ruditapes philippinarum* and *Mytilus galloprovincialis* (Pagliarani et al., 2006). Whether these enzymes exist in *L. pertusa* is unknown, but since corals possess a nervous system (Chen et al., 2008) and *L. pertusa* shows reaction to electrical stimulation (Shelton, 1980) at least the existence of $Na^+/K^+$-ATPase must be assumed. However, it remains unclear if this enzyme is participating in the modification of the calcifying fluid. The participation of Na/Ca ion pumps is also plausible, since it would result in higher Ca-concentrations in the calcifying space which would aid the calcification process due to the high transport capacity (Carafoli et al., 2001). Membrane calcium pumps on the other hand are better suited to transport Ca from a compartment with low Ca-concentrations, which is not applicable when considering seawater as the source compartment (Wang et al., 1992). Since the activity of enzymes is a function of temperature (Sizer, 2006), a temperature control of the ion concentration in the calcifying fluid has to be considered. Rising temperatures would increase the activity of the particular enzyme following the Arrhenius equation (Arrhenius, 1896) and consequently lower the Na-concentration in the calcifying space. Unfortunately, it is impossible to quantify these effects from the data at hand, because the optimum temperature and activation energy is not enzyme specific, but further controlled by enzyme and substrate purity and the presence of inhibitors or activators. Specific research is needed to identify the particular enzyme in the coral as well as determine the rate of ion-exchange although we note that an enzymatic control on aragonitic Na/Ca ratios does not necessarily imply a temperature control. In addition, besides a temperature control, there is also a pH control on enzymes (Trivedi and Danforth, 1966). While a positive correlation between Na/Ca and seawater pH is present in the samples utilized here,

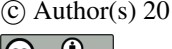



it is not possible to determine if this is caused by pH-controlled enzymatic activity or due to an
increased calcification rate. Higher seawater pH would cause higher calcification fluid pH which
would consequently also increase the aragonite saturation in the calcifying fluid (McCulloch et
al., 2012). The amount of pH up-regulation in the coral would therefore decrease, ultimately
conserving energy (≈ 10 % / -0.1 $pH_{SW}$) which can be used for ATP-dependent transport
proteins, pumping more Ca or $CO_3^{2-}$, leading to faster calcification (McCulloch et al., 2012).
The positive correlation between Na/Ca and pH might give more information about enzymes
that control the Na-concentration in the calcifying space. In foraminifera the existence of an
$Na^+/H^+$ exchanger has been discussed (Erez, 2003). Whether this exchanger exists in *L.*
*pertusa* as well, remains speculative but our data shows that it is unlikely to constitute the main
determining factor for the incorporation of sodium. If this would be the case there should be a
negative correlation between pH and Na/Ca ratio because in order to cope with lower pH-
values, the enzymatic activity would rise, pumping $H^+$ out of the calcifying space in exchange
for $Na^+$. Since there is a positive correlation, it can be concluded that either (1) *L. pertusa* does
not possess this type of ion exchange mechanism, (2) the effect of the $Na^+/H^+$-exchanger is
suppressed by other $Na^+$-pumping proteins ($Na^+/K^+$-ATPase, $Na^+/Ca^{2+}$-exchanger) or (3) the
process is overprinted by rate effects controlled by temperature or $[CO_3^{2-}]$.
Admittedly, the above discussion is only viable under the assumption of a closed calcifying
space with a much lower $[Na]_{ECF}$ than $[Na]_{Seawater}$. In the case of an open or semi-enclosed
calcifying space with $[Na]_{ECF}$ close or equal to $[Na]_{Seawater}$ the amount of Na removed by
enzymes or other ion-pumps is far too low to cause any significant changes in the composition
of the calcifying fluid with regards to Na. In combination with the low distribution coefficient,
changes in the Na-concentration of the ECF cannot cause the high variability of the skeletal
Na/Ca ratios. Since there is evidence for an at least semi-enclosed calcifying space (Tambutté
et al., 2011) we also consider this option. As described under Sec. 3.3 it is possible to calculate
the Mg-concentration of the ECF under the assumption of seawater leakage into the calcifying
space (Adkins et al., 2003; Gagnon et al., 2012) and a resulting approximately constant Na-





concentration. Based on this hypothesis, and the calculations defined in Eq. 3 and 4, we show
that the Mg-concentration in the ECF is constant, but with changing Ca-concentration (Fig. 7).
There is a large degree of scatter in the $[Mg]_{ECF}$ reconstructions (Fig. 7), which we suggest is
unlikely to represent real changes in the ECF [Mg] as it is difficult to envisage a purpose for
elevating $[Mg]_{ECF}$ above the of seawater given that it plays an inhibitory role in calcium
carbonate precipitation. It may be that the scatter above seawater values is derived from the
presence of organic material, as a small positive bias in measured coral Mg/Ca would result in
a large overestimation of $[Mg]_{ECF}$. Crucially however, we find that $[Mg]_{ECF}$ does not change as
a function of $[Ca]_{ECF}$, with the implication that in this model changing skeletal Mg/Ca and Na/Ca
ratios are not caused by changes of the Mg or Na-concentration of the ECF but rather are
entirely explicable through changes in the Ca-concentration. Again, this might be caused by
temperature-dependent enzyme or ion-pump activity but the affected pathway may be the
$Na^+/Ca^{2+}$-exchanger or $Ca^{2+}$-ATPase. Higher temperatures would then cause a higher
exchange capacity (Elias et al., 2001), leading to higher Ca- (Fig. 7) and lower Na-
concentrations in the ECF and consequent lower Mg/Ca and Na/Ca ratios. An elevation of [Ca]
in the ECF and the calcifying front is also supported by recent studies from Decarlo et al.,
(2018) and Sevilgen et al., (2019), who conducted Raman spectroscopic, $\delta^{11}B$ and
microsensor measurements on *Pocillopora damicornis, Acropora youngei and Stylophora*
*pistilla*. The results furthermore indicate the involvement of transcellular pathways to elevate
the Ca-concentration in the ECF (Sevilgen et al., 2019). The existence of $Na^+/Ca^{2+}$-exchangers
at least in warm-water corals is also supported by a recent study from Barron et al., (2018).
They gave evidence for the existence of $AyNCX_A$ exchangers and orthologous proteins, which
are very similar to $Na^+/Ca^{2+}$-exchangers known from vertebrates in all four tissue layers of
*Acropora yongei* and at least nine other coral species (Barron et al., 2018). The relative high
abundance in the calicoblastic layer suggests that these proteins fulfill a vital role in the
calcification process (Barron et al., 2018). The consistency of the concentration of this protein
with the occurrence of intracellular vesicles, possibly containing ACC (Mass et al., 2017) and
fusing with calicoblastic cells furthermore indicates processes of intracellular calcification



(Barron et al., 2018; Bertucci et al., 2011; Mass et al., 2013, 2014).While the existence of ACC
in corals is still debated (Akiva et al., 2018; DeCarlo, 2018; DeCarlo et al., 2018; Von Euw et
al., 2017), the process of intracellular calcification would also explain the resilience of corals
concerning environmental changes in pH and [$CO_3^-$] (Von Euw et al., 2017; McCulloch et al.,
2012). Intracellular calcification would also be beneficial to the former mentioned model of Na
pumping because the composition of the ECF and the surrounding seawater would then be
independent from the composition of the vesicles in which the calcification happens.
Even though a clear correlation between temperature and Na/Ca is present, the usefulness of
Na/Ca ratios is greatly reduced due to the large intraspecies variability. At 6°C Na/Ca ratios
vary by up to 20% and even up to 10 % in a single polyp. There are several reasons for this
great variability. One reason is the insufficient removal of the COC during the sampling
process. Due to the high growth rate and high organic content in the COC, elements, such as
Mg, Na and Li are enriched whereas other like U are depleted (Gagnon et al., 2007; Raddatz
et al., 2013, 2014b; Rollion-Bard and Blamart, 2015). This effect would also explain the high
Na/Ca values in corals from the Mediterranean Sea (T=13.56°C). It is possible that during the
sampling process a larger amount of the fibrous deposits was removed in comparison to the
other samples. This would cause a greater effect of the enriched COC material and therefore
cause higher Na/Ca ratios. It is therefore preferable to use laser ablation instead of solution-
based chemistry and profile measurements through the theca wall instead of bulk samples,
because it allows for a better recognition and removal of values that derive from COC or COC-
like structures. Seasonality could be also a factor responsible for a percentage of the variation,
but the sampled corals origin from depths where seasonality presumably only plays a minor
role. An estimated seasonal temperature change of 4°C only suffices to explain 1 mmol/mol
variation but not the visible variation of 10 mmol/mol. Inferring from this, there must be other
controls on Na/Ca ratios besides water temperature. Diurnal temperature fluctuations caused
by internal waves as found for example in the Rockall Through are also not high enough (3°C)
to explain these variations (Mienis et al., 2007). As mentioned under Sec 4.1, calcification rates
constitute a major control on Na/Ca ratios by controlling the amount of incorporation sites for



Na (Kitano et al., 1975; Mucci, 1988; White, 1977; Yoshimura et al., 2017). Therefore,
numerous second order control factors can cause variations of the Na/Ca ratios by controlling
the calcification rate. These second order controls include nutrient availability and supply,
changes in the carbonate system, coral fitness and many more. Some of these controls
(nutrient supply, coral fitness) have the potential to vary with a high spatial resolution and
consequently cause great variations in Na/Ca ratios even if the samples derive from the same
colony.
**4.3. Na/Mg ratios to overcome vital effects**
Even though a good correlation of $R^2$=0.9 between Na/Ca and temperature is observable in
our data, the samples from the Mediterranean Sea (T=13.54°C) show slightly elevated Na/Ca
ratios. Reasons for this are discussed in the prior chapter. Rollion-Bard and Blamart (2015)
proposed Na/Mg ratios to overcome these effects. This is possible because Na/Ca and Mg/Ca
ratios are controlled by similar vital effects such as growth rate and the amount of organic
content. Since there is no temperature effect on Mg/Ca ratios, by normalizing Na/Ca ratios to
Mg/Ca ratios, the impact of these vital effects on the calibration is greatly reduced (Fig. 8).
Regression for the Na/Mg – temperature correlation equals:
$$f_{T\ 6-22°C} = 7.1\ \pm 0.17 - 0.07 \pm 0.01 \times T\ (R^2 = 0.92, P = 0.009) \tag{5}$$
The application of Na/Mg in this study does not really improve the regression, as it removes
the inverse correlation between 6 and 10°C. This might be caused by covariance between
sodium and magnesium. It was shown that magnesium in the parent solution reduces the
amount of incorporated sodium. Furthermore, sodium in aragonite seems to decrease the
amount of some metal incorporation (Okumura and Kitano, 1986). However, utilizing Mg/Na
ratios removes the striking irregularity at 13.54 °C. This further proves the explanation for the
diverging Na/Ca ratios and facilitates an easy way to overcome inconsistencies during the
sampling process. The large scatter, however, is not significantly reduced which implies further
vital effects that cannot be resolved with this technique. To overcome this the mean of at least
10 analyzed samples should be used to get reliable results. If these prerequisites are fulfilled,





Na/Mg and Na/Ca ratios allow for a reliable temperature reconstruction. Advantageous to
Li/Mg ratios are the missing species-specific vital effects. This could prove useful especially
for temperature reconstructions in deep time on organisms that are extinct today. In this case
the nearest living relative principle is used, which potentially introduces large errors. Further
research on different aragonitic and calcitic organisms is necessary to detect further species
that show the same temperature sensitivity. Possibly Na/Ca ratios show no species-specific
variations at all and can therefore be used on extinct species where proxy calibrations are not
possible.
**5. Conclusion**
The data at hand does not support the usability of Na/Ca in corals as a salinity proxy as
proposed by Wit et al., (2013) and Mezger et al., (2016) for biogenic calcite. While there is a
positive trend between Na/Ca and salinity when excluding data from the Red Sea, there is no
statistical significance as tested with a one-way variance analysis.
A significant inverse correlation between temperature and Na/Ca ratios is present, which
cannot be explained by a co-variance of temperature and salinity (e.g. Mezger et al., 2016).
Two additional species, *Porites* sp. (Mitsuguchi et al., 2001; Ramos et al., 2004) and *M. edulis*
(Lorens and Bender, 1980) fit in this regression too. The mechanism of sodium incorporation
therefore seems to work equivalent in these three species. We propose temperature-
dependent Na-ion or Ca-Ion transport proteins as the underlying mechanism to explain the
observable correlation. While the intraspecies and intraindividual variation is large, averages
are rather accurate. Na/Ca ratios might provide a temperature-proxy that is usable for a wide
variety of aragonitic organisms and maybe even calcitic organisms (e.g. Mezger et al., 2016).
As proposed by Rollion-Bard and Blamart (2015), Na/Mg ratios can be used to correct for
inconsistencies during the sampling process.
Further research is needed to identify possible involved enzymes as well as quantify the effect
of further parameters that possible control the amount of sodium incorporation like growth-
and/or calcification rate.



**Author contribution**


Jacek Raddatz and Nicolai Schleinkofer designed the experiments and conducted the
measurements. Jacek Raddatz, Andre Freiwald, Lydia Beuck, Andres Rüggeberg and Volker
Liebetrau provided samples and environmental data. Nicolai Schleinkofer prepared the
manuscript with contributions from all co-authors.

**Acknowledgements**


We are grateful to all cruise captains, crew members and cruise participants of research
cruises POS325, POS391, POS455, POS 385, M61, POS625, B10-17a/b, 64PE284, M70/1,
COR2, MSM20-4, KRSE2013 and RV Gunnerus. Ship time of RV Belgica was provided by
BELSPO and RBINS–OD Nature. Cruise POS391 was realized by DFG Project RI 598/4-1.
JR acknowledges funding from DFG project ECHO RA 2516-1.
















**Figures**

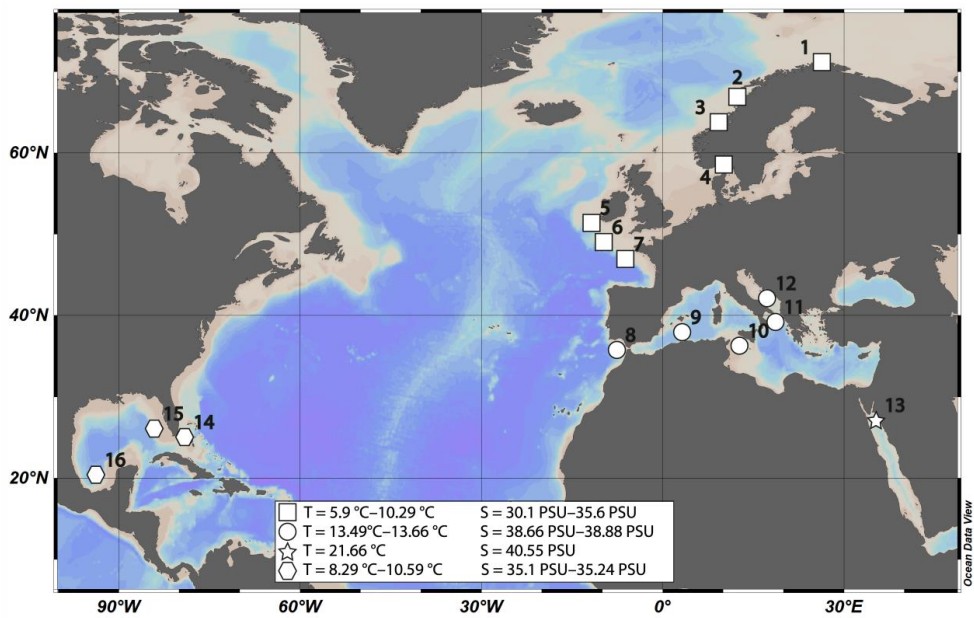


**Figure 1 Map of sampling locations. Locations are grouped in five areas with similar physical parameters.**
**1: Lopphavet, Sotbakken, Stjernsund; 2: Traenadjupet; 3: Sula, Nordleksa, Tautra, Røberg; 4: Oslofjord; 5:**
**Galway Mound, 6: Whittard Canyon; 7: Guilvinec Canyon; 8: Meknes Carbonate Mound Provinence 9: El**
**Idrissi Bank; 10: Urania Bank; 11: SML Provinence, 12: Bari Canyon; 13: Red Sea; 14: Great Bahama Bank;**
**15: Southwest Florida; 16: Campeche Bank**

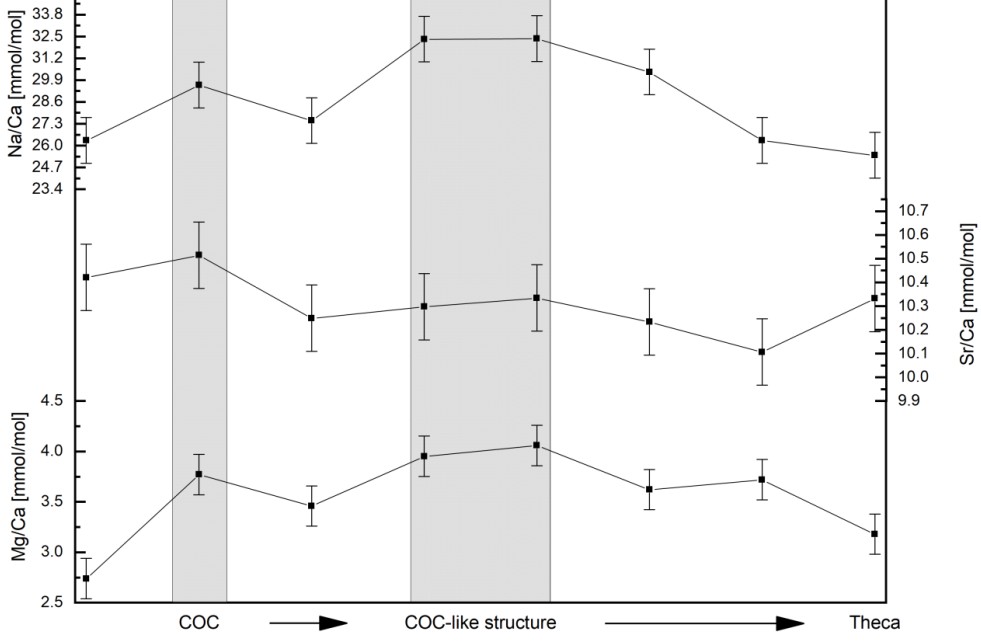


**Figure 2 Intra-individual element heterogeneities of one sample from Lopphavet (*L. pertusa*). Shaded-grey**
**areas indicate COC and COC-like structures. Error bars indicate 2SD of the JCp-1 mean. Within the**





uncertainty Sr/Ca ratios show no significant changes throughout the coral, whereas Mg/Ca and Na/Ca show
variations of 1.25 mmol/mol and 6 mmol/mol respectively.

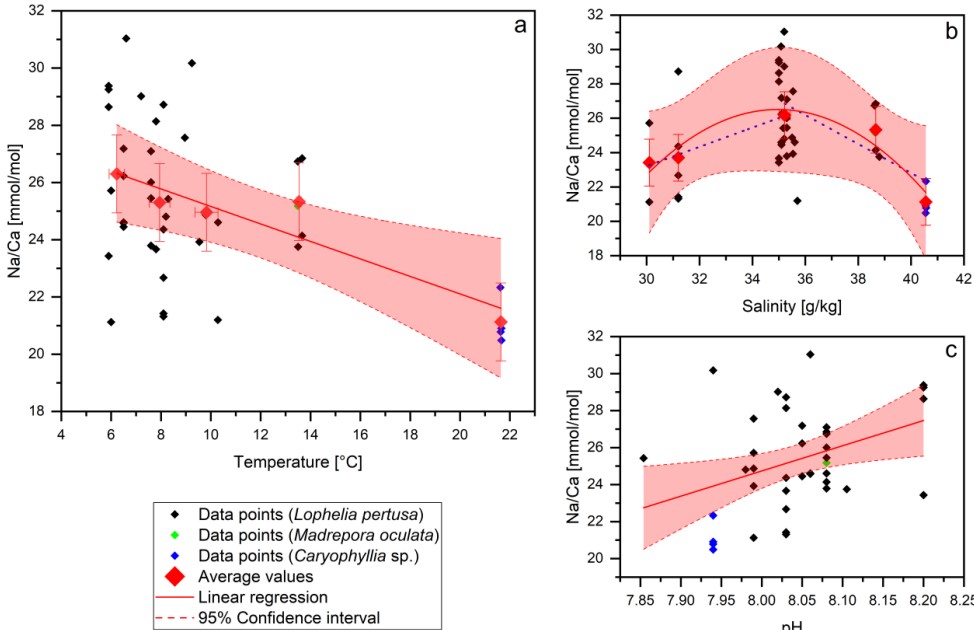


**Figure 3 Na/Ca Data (without COC) plotted against water temperature, salinity and pH. Red diamonds
indicate averaged values for temperature ranges. Temperature ranges are 5–7°C, 7–9°C, 9–11°C, 13–15°C
and 21–23°C. X-Error relates to the SD of the temperature/salinity mean. Y- Error bars indicate 2SD of the
JCp-1 mean. Red lines are linear regressions of the averaged values with the 95 % confidence interval
shaded. Blue dotted lines indicate linear regressions for different salinity ranges.**



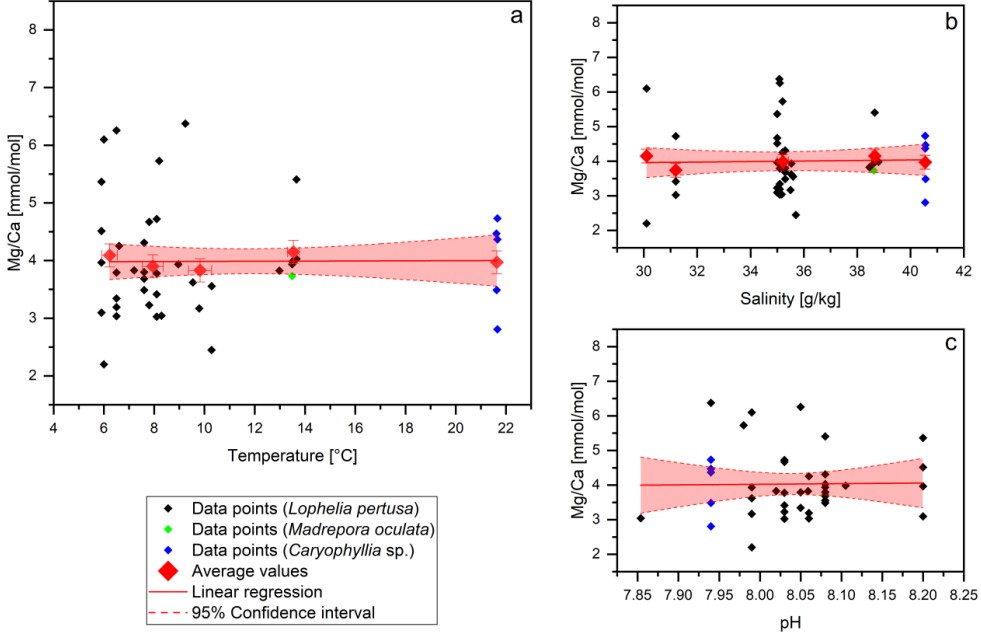

648

**Figure 4** Mg/Ca Data (without COC) plotted against water temperature, salinity and pH. Red diamonds
indicate averaged values for temperature ranges. Temperature ranges are 5–7°C, 7–9°C, 9–11°C, 13–15°C
and 21–23°C. X-Error relates to the SD of the temperature/salinity mean Y- Error bars indicate 2SD of the
JCp-1 mean. Red lines are linear regressions of the averaged values with the 95 % confidence interval
shaded.

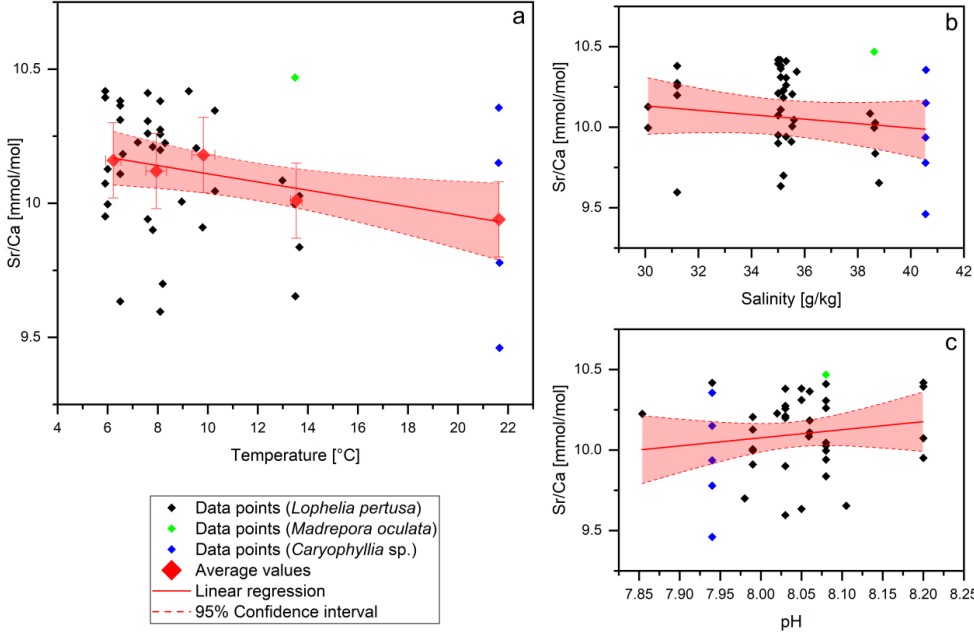


**Figure 5** Sr/Ca Data (without COC) plotted against water temperature, salinity and pH. Red diamonds
indicate averaged values for temperature ranges. Temperature ranges are 5–7°C, 7–9°C, 9–11°C, 13–15°C
and 21–23°C. X-Error relates to the SD of the temperature/salinity mean. Y- Error bars indicate 2SD of the



**JCp-1 mean. Red lines are linear regressions of the averaged values with the 95 % confidence interval**
**shaded.**

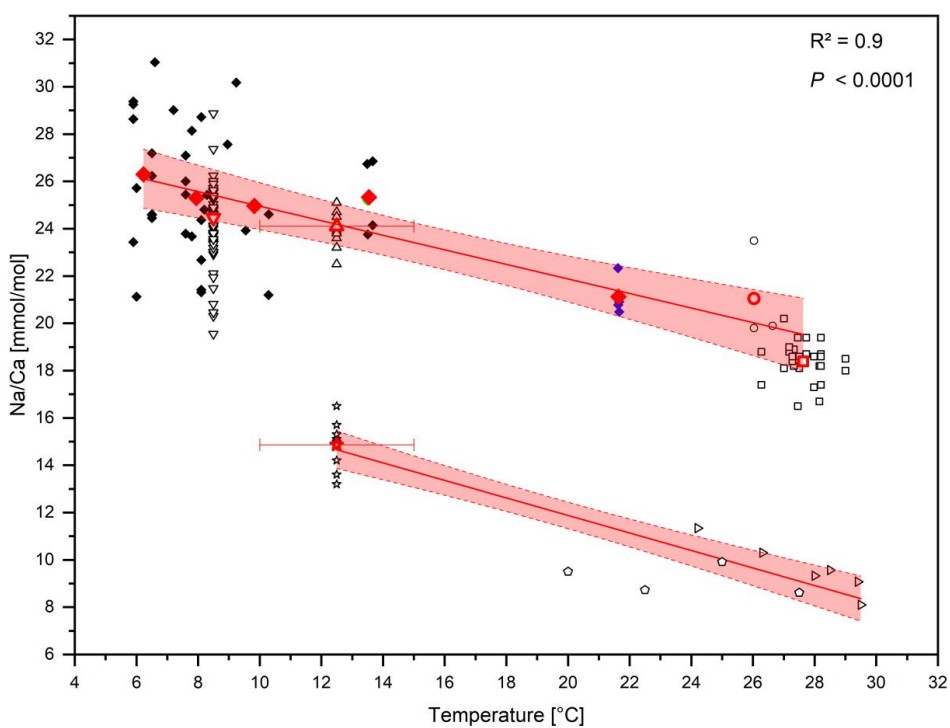

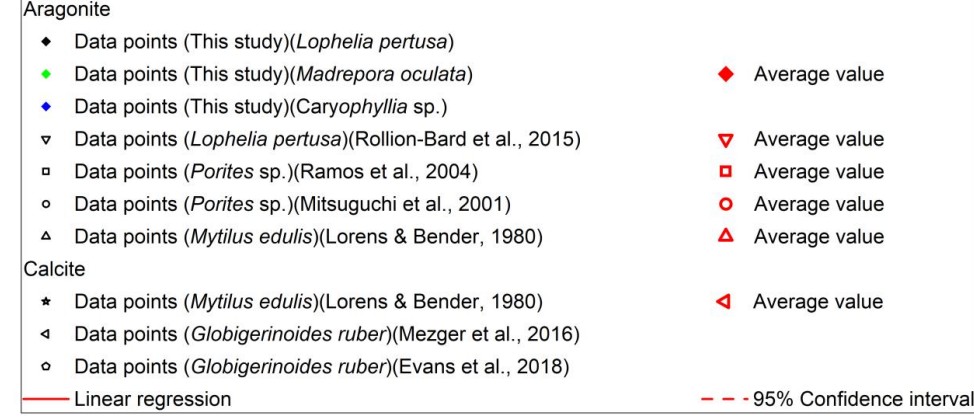

**Figure 6 Compiled Na/Ca ratios from different studies.** *L. pertusa, M. oculata, M. edulis* **and** *Porites* **sp. show**
**a negative linear relation with water temperature. R² relates only to the aragonitic samples Calcitic samples**
**from** *M. edulis* **and** *Globigerinoides ruber* **show the same sensitivity, albeit with an offset of 10 mmol/mol.**
**Temperature for the data from Lorens & Bender amounts to the average temperature of the tank the corals**
**were cultivated in while the error bars show maximum and minimum values.**







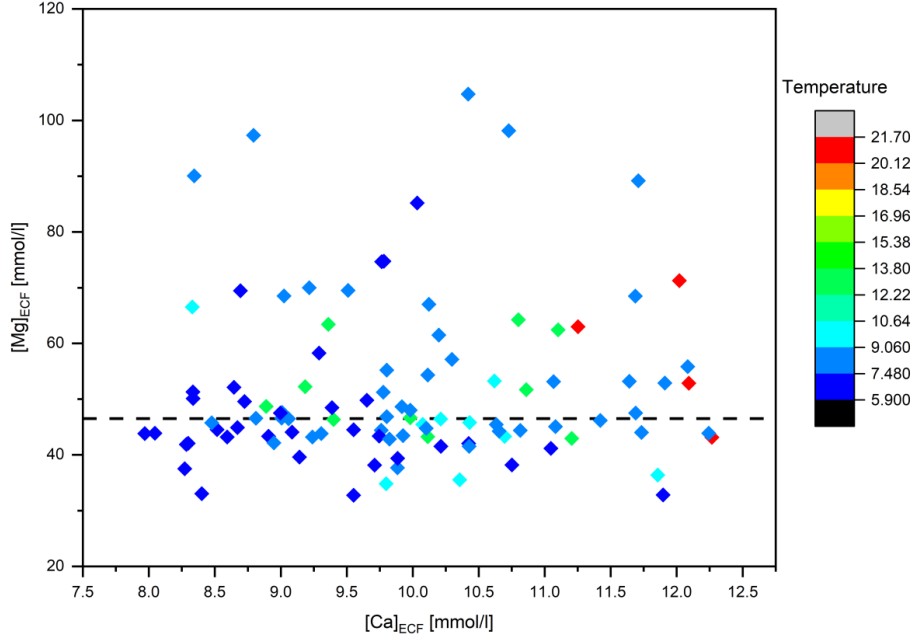


**Figure 7 Calcium and Magnesium concentration in the ECF of the investigated corals. The color of the data points indicate the ambient water temperature, which is increasing with increasing Ca-concentrations. The dashed line indicates the median of the Mg-concentration in the ECF.**




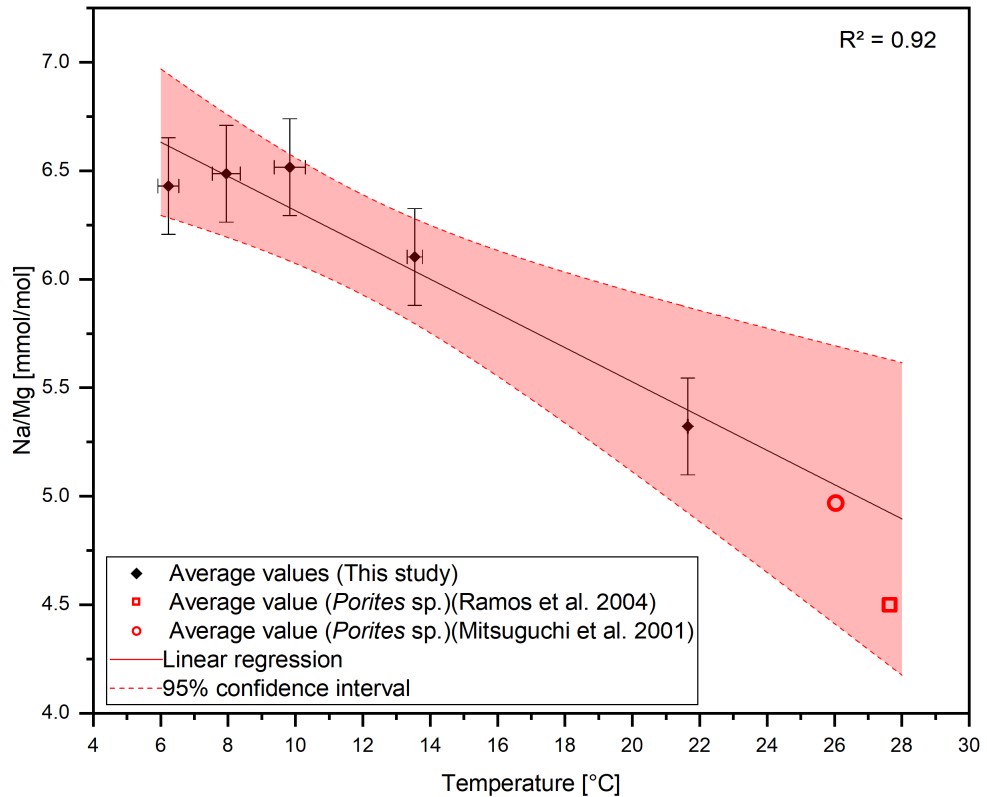

**Figure 8 Na/Mg ratios from this study vs. water temperature. Na/Mg ratios can be used to correct for the sampling of varying proportions of different domains. Y-Error bars relate to 2SD of the JCp-1 measurements. X-Error bars relate to 1SD of the temperature mean for the chosen temperature ranges.**





**Tables**

| Temperature [°C] | Na/Ca | | | Sr/Ca | | | Mg/Ca | | |
|---|---|---|---|---|---|---|---|---|---|
| | mmol/mol | n | SD | mmol/mol | n | SD | mmol/mol | n | SD |
| 6.23 ± 0.31 | 26.30 | 12 | 2.88 | 10.16 | 12 | 0.23 | 4.09 | 12 | 1.27 |
| 7.94 ± 0.41 | 25.30 | 15 | 2.48 | 10.13 | 15 | 0.24 | 3.90 | 14 | 0.74 |
| 9.83 ± 0.46 | 24.96 | 5 | 3.26 | 10.18 | 5 | 0.21 | 3.83 | 5 | 1.49 |
| 13.56 ± 0.09 (Na/Ca)/ 13.46 ± 0.25 (Mg/Ca, Sr/Ca) | 25.33 | 5 | 1.43 | 10.01 | 6 | 0.27 | 4.15 | 6 | 0.62 |
| 21.64 ± 0.02 | 21.13 | 4 | 0.82 | 9.94 | 5 | 0.34 | 3.97 | 5 | 0.8 |

| Average Salinity [g/kg] | Na/Ca | | | Sr/Ca | | | Mg/Ca | | |
|---|---|---|---|---|---|---|---|---|---|
| | mmol/mol | n | SD | mmol/mol | n | SD | mmol/mol | n | SD |
| 30.1 | 23.42 | 2 | 2.25 | 10.06 | 2 | 0.09 | 4.15 | 2 | 2.75 |
| 31.2 | 23.70 | 5 | 3.06 | 10.14 | 5 | 0.31 | 3.74 | 4 | 0.73 |
| 35.22 ± 0.21 | 26.18 | 25 | 2.46 | 10.16 | 25 | 0.22 | 3.99 | 25 | 1.01 |
| 38.67 ± 0.07 (Na/Ca)/ 38.64 ± 0.11 (Mg/Ca, Sr/Ca) | 25.33 | 5 | 1.43 | 10.01 | 6 | 0.27 | 4.16 | 6 | 0.62 |
| 40.56 ± 0.01 / 0.009 (Mg/Ca, Sr/Ca) | 21.13 | 4 | 0.82 | 9.94 | 5 | 0.34 | 3.97 | 5 | 0.8 |

**Table 1 Na/Ca, Sr/Ca, Mg/Ca mean values measured with ICP-OES, standard deviation and sample number.**
**Values relate to certain salinity and temperature envelopes.**

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
