# Peer review of "Environmental and biological controls on Na/Ca ratios in # scleractinian cold-water corals"

_Biogeosciences, 2019_

## Referee Comment (RC1) · Anonymous Referee #1 · 11 Mar 2019

The manuscript by Schleinkofer et al reports Na/Ca, Mg/Ca and Sr/Ca measurements in three cold-water corals. Their variations (mainly for Na/Ca) are discussed in view of the potential environmental and biological controls. Globally, the manuscript is well-written and easy to follow. Nevertheless, I think that the text could be shorten particularly in the introduction and improve in some explanations.

My comments are the following: -L23: please add the error - L48-63: I am not convinced that this part is really useful for the rest of the study. - L142: Specify here that the study of Branson et al (2015) is on foraminifera - L151: How this "0.18 mmol/mol" is calculated? Or is it measured? - L172: Why the measured Sr/Ca ratio is clearly higher

than the admitted ratio? - L174: Why do you mean by "accuracy amounts"? How was it calculated? - L189: Please define 'COC-like' - L190: Is there any relation between the increase and the species? - L200: Please indicate the errors on the measurements - L211: "As the P-values [...] in all these regressions." I do not understand this sentence. Please explain. - L222-224: the temperature is given with too many significant figures - L225: "Inorganic distribution coefficient is". Please correct. Please specify the temperature for the inorganic coefficient. - L234: Please add the errors - L239: There is only one study of Mg/Ca ratio in L. pertusa? - L247: same remark - L252: Please add the errors. One dot has to be removed in this sentence. - L255: Please add the errors - L257: Please correct the title - L267: Is there any reference for the influence of kinetics on Na? - L268: Is there any evidence for this concentration in the ECF? If there are some Ca2+/H+ pumps as stated by some authors, it would change the [Ca2+] of the ECF. - L274-275: I do not understand. As it was explained before, you constrained your calculation to have a mean [Ca] of $\approx$10 mmol/mol. So of course, the calculations will give a mean [Ca] close to 10 mmol/mol. Could you please better explain your point here? There is something that I do not understand in all these calculations. The Kd of corals is determined from the measurements in corals, divided by the concentration of the elements in seawater. So I do not see how you can calculate after that the [X] of the ECF. As an example, for Mg, the mean is the concentration of seawater, as we could expect from this calculation. You could perhaps try to calculate the concentrations in the ECF by assuming the partition coefficients of inorganic aragonite. - L308: I do not agree about this elevation of pH in the COC as the d11B values are lower in the COC than in fibres. - L312: This combination of different compartments with kinetic effects was already proposed in Meibom et al (2008) and Rollion-Bard et al (2010). - L336-338: Are these kinetics effects higher for Mg than for Sr? - L354-355: Please remove the sentence about foraminifera - L363-365: Is there any optimum of growth with T and/or pH? If yes, why is it not detectable on the relationships of Na/Ca and Mg/Ca with these parameters? If there is an effect of salinity on growth rates, why it is not observed in the relationship of Mg/Ca and salinity? - L382-388: In these studies, what is

the difference in Sr/Ca ratios between COC and fibres? Why the possible contribution of COC could be problematic for Sr/Ca and not for Mg/Ca and Na/Ca? - L445-L447: Please be consistent in the writing of Na+/K+-ATPase - L487-488: Previously some calculations were done with [Na]=455 mmol/mol. So what are the implications of a much lower [Na]ECF is your previous calculations? - L494-496: I am not convinced by the calculation of the Mg concentration in the ECF as I explained above - L501-502: As far as I know, Mg has an inhibitory role in the precipitation of calcite and not aragonite. Could you please add some references and more explain? - L514: "and" is in italic. Please correct. - L536-537: Please add references to Robinson et al (2014) and Rollion-Bard and Blamart (2014) as these two studies reviewed the geochemical differences between COC and fibres - L542: Why only this in situ technique? Are other techniques like EPMA and SIMS not suitable? - L574: Na/Mg instead of Mg/Na to be consistent - L581: Could it be also easier to measure Na than Li? - L589: Please specify that it is for cold-water corals. For tropical corals, please cite the study of Swart (1981)

Figure 1: What is the significance of the different symbols? I do not see the five areas.

Figure 2: Please add a picture of the sample that was measured for the location of COC and COC-like

Figures 3c, 4c, 4b and 4c: Why the averages are not represented in these figures?

Figure 6: Rollion-Bard and Blamart (2015) instead of Rollion-Bard et al (2015)

Figure 8: Why the value of Rollion-Bard and Blamart (2015) is not reported here?

Table 1: I do not understand the two temperatures of the lines 4

Please add a Table with the entire dataset.

---

## Referee Comment (RC2) · William Gray (Referee) · 25 Jul 2019

Review of Schleinkofer et al 'Environmental and biological controls on Na/Ca ratios in scleractinian cold-water corals'

William Gray william.gray@lsce.ipsl.fr

The new cold water coral Na/Ca, Mg/Ca and Sr/Ca data presented by Schleinkofer et al are a welcome addition to the literature, and overall the authors do a nice job of assessing the environmental controls on these elements. My greatest concern is that given the relatively limited dataset, the wording around some of the results can

be a overly strong, and caveats need to be added relating to the limited number of data points, and the fact that some of regressions are really only driven by one or two data points. The discussion also gets a little creative regarding pumps and calcification mechanisms, and is far too long given the limitations of the dataset.

Crucial information is missing from the methods regarding how pH was measured (probe, photometric dye, ALK+DIC)? What scale is pH on? If it was measured by different methods/on different scales, where efforts made to homogenise the dataset?

This is important because, despite the limited range in pH in the dataset (and likely the large uncertainty in the measured pH values), you see a significant relationship between Na/Ca and pH.

Given that temperature also influencing Na in your dataset it would better to regress Na against T and pH in multiple regression i.e. Na/Ca $\sim$ f(T + pH).

If you say Na is a promising T proxy, then what about the pH effect? This needs to be elaborated on.

It is important to add a new plot showing the covariance between predictor variables (T,S, pH) in your dataset.

Given that coral distribution (and thus optimum growth rate) is discussed in relationship to seawater density, why not regress Na/Ca against seawater density? It would be interesting to see if the Na 'peak' around 35 PSU relates to the optimum habitat density.

Minor comments: Line 38: I would describe Na/Ca in forams as a 'potential' tool, rather than a 'promising' tool. The relationships seen between Na/Ca and S in different studies conducted on the same species can vary wildly.

Line 50: it is not clear what you mean here – at a global scale it is not correct to say density is mainly governed by salinity (compare surface of warm salty tropical atlantic to cold fresh north pacific)

Line 108: see first comment – give the wildly different relationship between Na/Ca and salinity in different studies of same species it is not accurate to say Na in forams is largely function of salinity.

Line 135: much more information on the pH measurement method is needed here. What scale is pH given on?

Line 159: why is it important it can measure both axially and radially?

Line 218: given that coral distribution discussed in relation to seawater density, why not plot (and regress) Na/Ca (and Mg/Ca and Sr/Ca) against density? Interesting to see if Na 'peak' at 35 PSU relates to density preference of corals.

Line 241: you need to add this is essentially driven by one data point at ∼21.5 oC

Line 233: correlation with pH very interesting given limited pH range (and likely large errors). Na/Ca should be regressed against T and pH in a multiple regression to account for both variables.

Line 363: try plotting against density

Line 400 and 401: typo on signs in sensitivities

Lines 422-599: given the limitations of the dataset, this section needs to be made much shorter

Line 567: there really isn't enough data to say this. . .

Line 580: it is not all clear what you mean by 'Advantageous to Li/Mg ratios are the missing species-specific vital effects.' – if you are saying there are not vital effects in Na/Ca, there simply isn't enough data to say this

Please also note the supplement to this comment:
https://www.biogeosciences-discuss.net/bg-2019-40/bg-2019-40-RC2-supplement.pdf

---

## Author Comment (AC1) · 15 Aug 2019

We would like to thank William Gray for their time and effort in reviewing our manuscript and providing constructive criticism. We are confident that we can revise our manuscript to satisfy all of the reviewer's questions.

Major comments: Crucial information is missing from the methods regarding how pH was measured (probe, photometric dye, ALK+DIC)? What scale is pH on? If it was measured by different methods/on different scales, where efforts made to homogenise the dataset?
Thank you for your helpful and constructive comments. We added information about the method of pH measurement (L120 – 125). We also added a short comment about the pH effect on Na/Ca which could complicate temperature reconstructions (L563). Interestingly Na/Mg ratios show no correlation with pH.

"The seawater carbonate system data such as pH was taken from the associated cruise report (Flögel et al., 2014) or in case of the Red Sea and the western Atlantic from Mezger et al., (2016) and CARINA. Flögel et al., 2014 used a WTW Multi 350i compact precision hand- held meter to determine pH (Flögel et al., 2014), pH in the Red Sea was calculated from DIC and TA, measured during PELAGIA 64PE158 (Mezger et al., 2016), using CO2SYS (Lewis and Wallace, 1998). pH values are reported using the total scale."

It is important to add a new plot showing the covariance between predictor variables (T,S, pH) in your dataset.

We did not add plots about covariance of predictor variables because we mentioned important covariances in the text (L209, L228). We added all the environmental data and measured data as a supplement table, so these plots can be recreated if necessary.

Given that coral distribution (and thus optimum growth rate) is discussed in relationship to seawater density, why not regress Na/Ca against seawater density? It would be interesting to see if the Na 'peak' around 35 PSU relates to the optimum habitat density. The density vs. E/Ca ratio plots are very similar to the salinity vs. E/Ca plots. In deed the maximum Na/Ca ratios are found around 1028.25 kg/m$^3$. However, we do not think that these plots provide valuable information within the context of the rest of the manuscript. Nevertheless, we added these plots to this document and they can be easily reconstructed from the data in the supplements

We would however, respectfully disagree that this is a limited dataset. With 45 specimens from 16 locations with a wide temperature and salinity range, it contributes a

substantial amount of Na, Mg and Sr data from CWCs.

Minor comments:

Reviewer 2: L38 I would describe Na/Ca in forams as a 'potential' tool, rather than a 'promising' tool. The relationships seen between Na/Ca and S in different studies conducted on the same species can vary wildly

Response: L39 You are right. Results are not clear enough to justify calling it promising. We changed the wording.

Reviewer 2: L50 it is not clear what you mean here – at a global scale it is not correct to say density is mainly governed by salinity (compare surface of warm salty tropical Atlantic to cold fresh north pacific)

Response: Thank you for that remark, this probably has to be clarified. Generally speaking, you are right. In the ocean, the main control on density is the temperature. However, in this dataset, we have a temperature range of 15°C which translates to density changes of $\approx$ 4 kg/m$^3$ (constant salinity) and a salinity range of 10 g/kg which translates to a density change of $\approx$ >5 kg/m$^3$ (constant temperature). Accordingly, in this data set, changes in salinity are more important than changes in temperature. Nevertheless, we deleted this part (L51).

Reviewer 2: L108 see first comment – give the wildly different relationship between Na/Ca and salinity in different studies of same species it is not accurate to say Na in forams is largely function of salinity

Response: We changed the statement (L98)

Reviewer 2: L135 much more information on the pH measurement method is needed here. What scale is pH given on?

Response: L126 We added more information about pH measurements. Values are given in pHT and data was not homogenized

[Figure]
Reviewer 2: L159: why is it important it can measure both axially and radially?

Response: Being capable of measuring both axially and radially is important when measuring alkali metals such as sodium, because these elements are better measured in radial view. Axial view is more affected by excitation disturbance (Ivaldi and Tyson, 1995) which especially influences the easy electron transitions (alkali metals) (Demers, 1979). However, trustworthy results are also possible with axial view (e.g. (Bertlich et al., 2018))

Reviewer 2: L218: given that coral distribution discussed in relation to seawater density, why not plot (and regress) Na/Ca (and Mg/Ca and Sr/Ca) against density? Interesting to see if Na 'peak' at 35 PSU relates to density preference of corals.

Response: Plots of seawater density vs. Mg,Na,Sr/Ca are very similar to the Mg,Na,Sr/Ca vs. salinity plots. Na/Ca vs. density shows a peak at a density of 1028.25 kg/m$^3$ which is close to the preferred values in the northern Atlantic (1027.35 – 1027.65 kg/m$^3$)(Dullo et al., 2008) and the Mediterranean sea (1029 kg/m$^3$)(Flögel et al., 2014). Seawater density was calculated with the formula given in Fofonoff and Millard, (1983) from temperature, salinity and depth. For Mg/Ca and Sr/Ca plots show no significant trends with changing density. (Fig. 1,2,3)

Reviewer 2: L241 you need to add this is essentially driven by one data point at âĹij21.5 oC

Response: L234 We added a remark that data availability for M. occulata and Caryophylliidae is too scarce to draw any firm conclusions. There might be a typo in your comment. You are probably referring to line 214 not 241. We added a comment on this as well (L213)

Reviewer 2: L233: correlation with pH very interesting given limited pH range (and likely large errors). Na/Ca should be regressed against T and pH in a multiple regression to account for both variables.

Response: Thank you for that comment. We did try a multiple linear regression, but the slope of the multiple regression for temperature is very similar to the single linear regression (-0.33 (multiple linear regression) vs. 0.31). Accordingly, pH does not seem to have a big effect on Na/Ca ratios. Since we can see a trend to lower pH values with increasing temperature the pH-effect on Na/Ca ratios might very well be just an effect of this covariation.

Reviewer 2: L363: try plotting against density

Response: See Major comments and plots

Reviewer 2: L400 and L401: typo on signs in sensitivities

Response: L410 Thank you, we changed it to minus (-)

Reviewer 2: L422-599: given the limitations of the dataset, this section needs to be made much shorter

Response: We deleted lines 476-487 and 516 – 531 but we do not think that we can shorten the section more without deleting important information.

Reviewer 2: L567: there really isn't enough data to say this...

Response: Thank you for that comment. In deed it might be a little bit overzealous to speak of greatly reduced vital effects but the data clearly shows that there are effects that could complicate temperature reconstruction with Na/Ca ratios, which however are minimized when using Na/Mg ratios. We adjusted the wording (L552)

Reviewer 2: L580: it is not all clear what you mean by 'Advantageous to Li/Mg ratios are the missing species-specific vital effects.' – if you are saying there are not vital effects in Na/Ca, there simply isn't enough data to say this

Response: L574 You are right, we deleted the statement

Changes are marked green in the Manuscript

Additional changes: changed Lophelia pertusa to Desmophyllum pertusum (new accepted species name)

[Figure]

**Fig. 1.** Na/Ca vs. density

[Figure]

**Fig. 2.** Sr/Ca vs. density

[Figure]

Fig. 3. Mg/Ca vs. density

---

## Author Comment (AC2) · 15 Aug 2019

We thank Reviewer 1 for providing very constructive comments, which considerable improved our manuscript. We are confident that we have revised our manuscript such that all the reviewer's questions have been addressed.

Reviewer 1: L23: please add the error

Response: L 24: Error added

Reviewer 1: L48-63: I am not convinced that this part is really useful for the rest of the study

[Figure]

Response: Thank you for pointing that out. In deed it does not add much valuable information. We deleted this part except for the section about density control on the spatial distribution because it is closely related to salinities.

Reviewer 1: L142: Specify here that the study of Branson et al (2015) is on foraminifera

Response: L139: We added a clarification

Reviewer 1: L151: How this "0.18 mmol/mol" is calculated? Or is it measured?

Response: L147 Thank you for that comment, we agree a clarification was needed. This is calculated by using the profiled samples were the COC is recognizable. This value is just the difference between the means of the samples including COC measurements and the means of the samples not including the COC measurements.

Reviewer 1: L172: Why the measured Sr/Ca ratio is clearly higher than the admitted ratio?

Response: There could be many explanations for this error. One could be heterogeneities in the JcP-1 standard powder (Runnalls and Coleman, 2003). However, these deviations are accounted for and corrected during processing of the data.

Reviewer 1: L174: Why do you mean by "accuracy amounts"? How was it calculated?

Response: It is the deviation between the measured JCp-1 values and the true values in percent. We thought it is a convenient way to show the quality of measurements. We removed it nonetheless to avoid confusion.

Reviewer 1: L189: Please define 'COC-like'

Response: L185: We added a definition in the manuscript "In the COC and COC-like structures (structures that geochemically correspond to COC but morphologically to fibrous deposits)"

Reviewer 1: L190: Is there any relation between the increase and the species?

Response: We thank the reviewer for this very interesting comment, which should be targeted in future studies. However, here we cannot comment on this because the profiled samples all derive from Lophelia pertusa. However, it seems unlikely that great differences between different species exist given that the main controlling factor appears to be the increased growth rate in the COC's which most likely similar in different species. The main reason for the different increase in this study is probably sample mixing between COC material and fibrous material with different percentages.

Reviewer 1: L200: Please indicate the errors on the measurements

Response: L198,236,244 Added Error values

Reviewer 1: L211: "As the P-values [...] in all these regressions." I do not understand this sentence. Please explain.

Response: The P-values are the result of an ANOVA test for regression coefficients. Since the values are higher than the chosen confidence level (95%) the regressions have to be considered as non-significant.

Reviewer 1: L222-224: the temperature is given with too many significant figures

Response: L221-222 Reduced to one significant figure

Reviewer 1: L225: "Inorganic distribution coefficient is". Please correct. Please specify the temperature for the inorganic coefficient.

Response: L225 Corrected and added the temperature for the distribution coefficient (15°C)

Reviewer 1: L234: Please add the errors

Response: L227 Errors added

Reviewer 1: L239: There is only one study of Mg/Ca ratio in L. pertusa? - L247: same remark

Response: L239 & L247 We added more references to other studies

Reviewer 1: L252: Please add the errors. One dot has to be removed in this sentence

Response: L252 Errors added and comma removed

Reviewer 1: L255 Please add the errors

Response: L256 Errors added

Reviewer 1: L257 Please correct the title

Response: L258 Title corrected

Reviewer 1: L267 Is there any reference for the influence of kinetics on Na?

Response: L269 Reference added

Reviewer 1: L268: Is there any evidence for this concentration in the ECF? If there are some Ca2+/H+ pumps as stated by some authors, it would change the [Ca2+] of the ECF.

Response: Evidence for this concentration is given by the micro sensor studies on Galaxea fascicularis (Al-Horani et al., 2003, L278). This is exactly the point we are making, that changes in e.g Na/Ca are controlled by the [Ca2+] rather than [Na+], at least when assuming seawater leakage into the calcification site.

Reviewer 1: L274-275: I do not understand. As it was explained before, you constrained your calculation to have a mean [Ca] of ≈10 mmol/mol. So of course, the calculations will give a mean [Ca] close to 10 mmol/mol. Could you please better explain your point here? There is something that I do not understand in all these calculations. The Kd of corals is determined from the measurements in corals, divided by the concentration of the elements in seawater. So I do not see how you can calculate after that the [X] of the ECF. As an example, for Mg, the mean is the concentration of seawater, as we could expect from this calculation. You could perhaps try to calculate the concentrations in the ECF by assuming the partition coefficients of inorganic aragonite.

Response: Thank you for pointing that out, we refrained from using the inorganic distribution coefficient because it would result in much lower predicted Na/Ca ratios not to constrain a [Ca]ECF = 10. We corrected that. The calculations are based on the assumption that [Element]ECF is close to its concentration in seawater. The whole point here is to illustrate that changes in Mg/Ca ratios of the ECF and consequently in the aragonite are not caused by changes of [Mg] but by changes in [Ca]. Using partition coefficients from inorganic aragonite would decrease the calculated concentration but it would not change the general picture of an increasing [Ca]ECF with higher temperatures and relatively constant [Mg] values.

Reviewer 1: L308: I do not agree about this elevation of pH in the COC as the d11B values are lower in the COC than in fibres.

Response: This is true. U/Ca measurements on the other hand indicate an elevation at the COC (Raddatz et al., 2014; Sinclair et al., 2006). We added a clarification (L314) that the pH-elevation at COC is not finally resolved and also pH decreases are possible. Also, studies based on $\delta$11B measurements show that the COC might be an area of lower pH-values compared to the fibrous zones (Blamart et al., 2007; Jurikova et al., 2019; Rollion-Bard et al., 2011)

Reviewer 1: L312: This combination of different compartments with kinetic effects was already proposed in Meibom et al (2008) and Rollion-Bard et al (2010).

Response: L3312 Added reference

Reviewer 1: L336-338: Are these kinetics effects higher for Mg than for Sr? Response: The kinetic effects on Mg and Sr in coral aragonite are not resolved yet. The kinetic effects on Mg mentioned in line 336-338 were investigated on inorganically precipitated aragonite (clarification added). Given the available data and the data in this set, we

would conclude that the effect on Mg is stronger as well as doubt that there is a kinetic effect on Sr in the first place (Gabitov et al., 2006, 2008). Our Sr/Ca shows no regular covariance with the fast calcifying COC's and occurring covariances can be explained by the high organic content in the COC's. Mg/Ca on the other hand show regularly increasing values in the COC's. While this is also explainable with the occurrence of ACC or higher organic contents, a proportionate influence of calcification rate must be assumed, give the results from inorganic precipitation experiments. It is however, also possible that that the calcification rate control on Mg/Ca in corals is suppressed by other biological effects.

Reviewer 1: L354-355: Please remove the sentence about foraminifera

Response: L356 Sentence removed

Reviewer 1: L363-365: Is there any optimum of growth with T and/or pH? If yes, why is it not detectable on the relationships of Na/Ca and Mg/Ca with these parameters? If there is an effect of salinity on growth rates, why it is not observed in the relationship of Mg/Ca and salinity?

Response: There are no values known for optimum growth of cold-water corals. Considering the optimum pH, we would assume that a higher pH is beneficial for the growth rates (Büscher et al., 2017) which is also visible in Fig. 3 (higher pH → higher Na/Ca). In terms of temperature we would also assume that up to a certain threshold, higher temperatures benefit the corals growth (Büscher et al., 2017). This should then lead to higher Na/Ca values with higher temperatures, but it is very likely that this effect is just suppressed by the temperature effect as the growth rate changes introduced by different temperatures are far lower compared to the growth rate changes caused by the different skeletal compartments. Therefore Mg/Ca ratios might not be sensitive enough to show any changes introduced by the small growth rate changes.

Reviewer 1: L382-388: In these studies, what is the difference in Sr/Ca ratios between COC and fibres? Why the possible contribution of COC could be problematic for Sr/Ca

and not for Mg/Ca and Na/Ca?

Response: The differences range from +0.3 − +0.6 to − 0.8. The contribution of COC is also problematic for Mg/Ca and Na/Ca but it would not change the general trend. For Sr/Ca on the other hands it is also reported that the ratios decrease through the fibrous zones and then increase again in the COC (Gagnon et al., 2007). Depending on the exact drilling location and the consequent mixing of aragonite from different compartments, this could give the impression of decreasing Sr/Ca ratios in the COC, but it is just caused by the sample mixing.

Reviewer 1: L445-L447: Please be consistent in the writing of Na+/K+-ATPase

Response: L437,439,444 Corrected

Reviewer 1: L487-488: Previously some calculations were done with [Na]=455 mmol/mol. So what are the implications of a much lower [Na]ECF is your previous calculations?

Response: In case of a lower [Na]ECF the contribution of Ca transport systems is not necessary to explain the temperature sensitivity of Na/Ca ratios. In this case the coral would be able control the elemental composition of the ECF through Na transporting enzymes. This is not possible if [Na]ECF = 455 mmol/l because the effect of Na transporting enzymes would be negligible. Considering the calculations, using lower [Na]ECF values would also decrease the calculated [Ca]ECF and [Mg]ECF values but it would not change the general idea and result that [Mg]ECF concentrations are stable but [Ca]ECF changes.

Reviewer 1: L501-502: As far as I know, Mg has an inhibitory role in the precipitation of calcite and not aragonite. Could you please add some references and more explain?

Response: You are right in the pure chemical sense that Mg does not inhibit aragonite formation but there is evidence that it inhibits aragonite growth (Swart 1981) on a biological level because it acts antagonistic to the calcium transport(Okazaki, 1956;

Swart, 1981; Yamazato, 1966). We added an explanation (L492)

Reviewer 1 L514: "and" is in italic. Please correct

Response: L505 Corrected

Reviewer 1 L536-537: Please add references to Robinson et al (2014) and Rollion-Bard and Blamart (2014) as these two studies reviewed the geochemical differences between COC and fibres

Response: L514 References added

Reviewer 1 L542: Why only this in situ technique? Are other techniques like EPMA and SIMS not suitable?

Response: Other techniques are of course equally suitable. We added them to the Manuscript (L519)

Reviewer 1 L574: Na/Mg instead of Mg/Na to be consistent

Response: L550 Corrected

Reviewer 1 L581: Could it be also easier to measure Na than Li?

Response: L574 That's right, thank you for mentioning. The different abundance in the aragonite alone makes Na easier to measure than Li (15-30 mmol/mol Na/Ca, 10-20 $\mu$mol/mol Li/Ca). However, we deleted the sentence as the data availability does not allow to make assumptions about vital effects. Reviewer 1 L589: Please specify that it is for cold-water corals. For tropical corals, please cite the study of Swart (1981)

Response: Modified to clarify that it is for cold-water corals L567

Reviewer 1 Figure 1: What is the significance of the different symbols? I do not see the five areas

Response: Thank you for pointing that out. There are only 4 different areas, which should give the reader a fast overview over the different environmental parameters.

Reviewer 1 Figure 2: Please add a picture of the sample that was measured for the location of COC and COC-like

Response: L628 We added a picture

Reviewer 1 Figures 3c, 4c, 4b and 4c: Why the averages are not represented in these figures?

Response: Averages added to 5b. While the averages in the temperature/salinity plots do help to make certain characteristics better visible, we think they do not improve the pH plots. However, we added them to the graphs (L634,640,646).

Reviewer 1 Figure 6: Rollion-Bard and Blamart (2015) instead of Rollion-Bard et al (2015)

Response: Corrected (L653)

Reviewer 1 Figure 8: Why the value of Rollion-Bard and Blamart (2015) is not reported here?

Response: Thank you for mentioning. We added these values as well as values from Swart 1981 (L665)

Reviewer 1 Table 1: I do not understand the two temperatures of the lines 4

Response: The two temperatures are caused by outliers. In this temperature range e.g. the Mg/Ca of sample 1 was an outlier but not Na/Ca, in sample 2 Na/Ca had to be removed but not Mg/Ca. Mg/Ca values therefore relate to a slightly different mean temperature than Na/Ca or Sr/Ca values. However, as this is not the correct way to treat outliers we modified the table.

Reviewer 1 Please add a Table with the entire dataset.

Response: We added a table containing the entire data set

Changes are marked red in the manuscript

Additional Changes: changed Caryophyllia sp. to Caryophylliidae

References: Al-Horani, F. A., Al-Moghrabi, S. M. and De Beer, D.: The mechanism of calcification and its relation to photosynthesis and respiration in the scleractinian coral Galaxea fascicularis, Mar. Biol., 142(3), 419–426, doi:10.1007/s00227-002-0981-8, 2003.

Büscher, J. V., Form, A. U. and Riebesell, U.: Interactive Effects of Ocean Acidification and Warming on Growth, Fitness and Survival of the Cold-Water Coral Lophelia pertusa under Different Food Availabilities, Front. Mar. Sci., 4(April), 1–14, doi:10.3389/fmars.2017.00101, 2017.

Gabitov, R. I., Cohen, A. L., Gaetani, G. A., Holcomb, M. and Watson, E. B.: The impact of crystal growth rate on element ratios in aragonite: An experimental approach to understanding vital effects, Geochim. Cosmochim. Acta, 70(18), A187, doi:10.1016/j.gca.2006.06.377, 2006.

Gabitov, R. I., Gaetani, G. A., Watson, E. B., Cohen, A. L. and Ehrlich, H. L.: Experimental determination of growth rate effect on $U^{6+}$ and $Mg^{2+}$ partitioning between aragonite and fluid at elevated $U^{6+}$ concentration, Geochim. Cosmochim. Acta, 72(16), 4058–4068, doi:10.1016/j.gca.2008.05.047, 2008.

Gagnon, A. C., Adkins, J. F., Fernandez, D. P. and Robinson, L. F.: Sr/Ca and Mg/Ca vital effects correlated with skeletal architecture in a scleractinian deep-sea coral and the role of Rayleigh fractionation, Earth Planet. Sci. Lett., 261(1–2), 280–295, doi:10.1016/j.epsl.2007.07.013, 2007.

Okazaki, K.: SKELETON FORMATION OF SEA URCHIN LARVAE. I. EFFECT OF CA CONCENTRATION OF THE MEDIUM, Biol. Bull., 110(3), 320–333, doi:10.2307/1538838, 1956.

Raddatz, J., Rüggeberg, A., Flögel, S., Hathorne, E. C., Liebetrau, V., Eisenhauer, A. and Dullo, W. C.: The influence of seawater pH on U/Ca ratios in the scleractinian

cold-water coral Lophelia pertusa, Biogeosciences, 11(7), 1863–1871, doi:10.5194/bg-11-1863-2014, 2014.

Runnalls, L. A. and Coleman, M. L.: Record of natural and anthropogenic changes in reef environments (Barbados West Indies) using laser ablation ICP-MS and sclerochronology on coral cores, Coral Reefs, 22(4), 416–426, doi:10.1007/s00338-003-0349-7, 2003.

Sinclair, D. J., Williams, B. and Risk, M.: A biological origin for climate signals in corals - Trace element "vital effects" are ubiquitous in Scleractinian coral skeletons, Geophys. Res. Lett., 33(17), 1–5, doi:10.1029/2006GL027183, 2006.

Swart, P. K.: The strontium, magnesium and sodium composition of recent scleractinian coral skeletons as standards for palaeoenvironmental analysis, Palaeogeogr. Palaeoclimatol. Palaeoecol., 34(C), 115–136, doi:10.1016/0031-0182(81)90060-2, 1981.

Yamazato, K.: Calcification in a solitary coral, Fungia scutaria Lamarck in relation to environmental factors, 1966.

Please also note the supplement to this comment:
https://www.biogeosciences-discuss.net/bg-2019-40/bg-2019-40-AC2-supplement.pdf

[Figure]

**Fig. 1.** Picture of used sample with measured tracks

**Supplement:**

| Samplename | Location | Coordinates | Waterdepth [m] | Temperature [°C] | Salinity [PSU] | pH | Reference |
|---|---|---|---|---|---|---|---|
| A1 - A4 | Traenadjupet, Norwegian Sea | 66°58.400'N 11°06.529'E | 300 | 7.2 | 35.2 | 8.02 | Dullo et al. (2008), Rüggeberg et al. (2011) |
| B1 - B2 | Whittard Canyon, Celtic Sea | 48°46.79'N 10°34.20'W | 835 | 9.79 | 35.5 | 7.99 | Raddatz et al. (2013) |
| C1 - C2 | Stjernsund Fjord, Norwegian Sea | 70°16.04'N 22°27.37'E | 295 | 5.9 | 35 | 8.20 | Rüggeberg et al. (2011) |
| D1 – D4 | Little Galway Mound, Belgica Mound Province | 51°26.51'N 11°45.43'W | 881 | 8.96 | 35.53 | 7.99 | Dullo et al. (2008), Rüggeberg et al. (2011) |
| E1 - E3 | Santa Maria di Leuca, Ionic Sea | 39°125.00'N 18°127.00'E | 560–750 | 13.66 | 38.66 | 8.08 | Flögel et al. (2014) |
| F1 – F4 | Guilvinec Canyon, Bay of Biscay | 46°56.20'N 5°21.60'W | 800 | 10.29 | 35.6 | 8.08 | Raddatz et al. (2013) |
| G1 – G4 | Sotbakken, Norwegian Sea | 70°45.35'N 18°40.04'E | 265 | 6.6 | 35.2 | 8.06 | Raddatz et al. (2013) |
| H1 – H3 | Galway Mound, Belgica Mound Province | 51°26.94'N 11°45.16'E | 837 | 9.54 | 35.53 | 7.99 | Dullo et al. (2008), Rüggeberg et al. (2011) |
| I1 – I3 | Urania Bank, Strait of Sicily | 36°50.32'N 13°09.35'E | 559 | 13.5 | 38.8 | 8.11 | Raddatz et al. (2013), López Correa et al. (2010) |
| J1 – J2 | Meknes Carbonate Mound Provinence, Gulf of Cadiz | 34.59.98'N 7°04.51'W | 738 | 10.28 | 35.7 | 7.76 | Raddatz et al. (2013) |
| K1 – K5 | Trondheimfjord, Norwegian Sea | 63°28.61'N 9°59.72'E | 240 | 8.1 | 31.2 | 8.03 | Dullo et al. (2008), Rüggeberg et al. (2011) |
| LO1 | Bari Canyon, Adriatic Sea | 41°18.00'N 17°12.00'E | 315–650 | 13.49 | 38.62 | 8.08 | Flögel et al. (2014) |
| MSS1 | Santa Maria di Leuca, Ionic Sea | 39°25.00'N 18°27.00'E | 560–750 | 13.66 | 38.66 | 8.08 | Flögel et al. (2014) |
| MA1 | Bari Canyon, Adriatic Sea | 41°18.00'N 17°12.00'E | 315–650 | 13.49 | 38.62 | 8.08 | Flögel et al. (2014) |

| | | | | | | |
|---|---|---|---|---|---|---|
| MSM1 | Campeche Bank, Gulf of Mexico | 23°50.121'N 87°10.484'W | 565 | 10.59 | 35.24 | 7.94 | WOA 2013 (Locarnini et al., 2013; Zweng et al., 2013) |
| MSM2 | Great Bahama Bank Mound E, Northwest Atlantic | 24°35.623'N 79°16.808'W | 579 | 9.24 | 35.084 | 7.94 | WOA 2013 (Locarnini et al., 2013; Zweng et al., 2013) |
| MSM3 | Great Bahama Bank Mound B, Northwest Atlantic | 24°33.848'N 79°19.811'W | 616 | 9.27 | 35.1 | 7.94 | WOA 2013 (Locarnini et al., 2013; Zweng et al., 2013) |
| MSM4 | Southwest Florida, Gulf of Mexico | 24°58.164'N 84°17.973'W | 483 | 8.29 | 35.17 | 7.85 | WOA 2013 (Locarnini et al., 2013; Zweng et al., 2013) |
| ALB1 | El Idrissi Bank, Alboran Sea | 36°06.31'N 3°29.33'W | 647 | 12.98 | 38.464 | 8.06 | WOA 2013 (Locarnini et al., 2013; Zweng et al., 2013) |
| KRS1 | Northeastern Red Sea | 27°42.306'N 35°09.046'E | 954 | 21.66 | 40.55 | 7.94 | WOA 2013 (Locarnini et al., 2013; Zweng et al., 2013) |
| KRS2 | Northeastern Red Sea | 22°17.857'N 38°53.648'E | 580 | 21.66 | 40.55 | 7.94 | WOA 2013 (Locarnini et al., 2013; Zweng et al., 2013) |
| KRS3 | Northeastern Red Sea | 27°42.306'N 35°09.046'E | 954 | 21.66 | 40.55 | 7.94 | WOA 2013 (Locarnini et al., 2013; Zweng et al., 2013) |
| KRS4 | Northeastern Red Sea | 27°42.306'N 35°09.046'E | 954 | 21.66 | 40.55 | 7.94 | WOA 2013 (Locarnini et al., 2013; Zweng et al., 2013) |
| KRS5 | Northeastern Red Sea | 22°46.149'N 38°02.944'E | 625 | 21.66 | 40.55 | 7.94 | WOA 2013 (Locarnini et al., 2013; Zweng et al., 2013) |
| L10-1 | Tautra Reef, Norwegian Sea | 63°35.36'N 10°31.23'E | 39 | 6 | 30.1 | 7.99 | Neulinger (2008) |
| L2-1 – L2-2 | Lopphavet, Norwegian Sea | 70°26.80'N 21°10.38'E | 230 | 6.5 | 35.1 | 8.06 | Raddatz et al. (2013) |

| L3-1 – L3-2 | Lopphavet, Norwegian Sea | 70°26.80'N 21°10.38'E | 230 | 6.5 | 35.1 | 8.06 | Raddatz et al. (2013) |
|---|---|---|---|---|---|---|---|
| L5-1 – L5-2 | Stjernsund Fjord, Norwegian Sea | 70°16.04'N 22°27.37'E | 295 | 5.9 | 35 | 8.20 | Rüggeberg et al. (2011) |
| L6-1 – L6-2 | Sula Reef, Norwegian Sea | 64°05.93'N 8°05.47'E | 286 | 7.6 | 35.3 | 8.08 | Raddatz et al. (2013) |
| L7-1 – L7-2 | Sula Reef, Norwegian Sea | 64°05.93'N 8°05.47'E | 286 | 7.6 | 35.3 | 8.08 | Raddatz et al. (2013) |
| L8-1 – L8-2 | Nordleksa, Norwegian Sea | 63°36.43'N 9°22.66'E | 155 | 7.8 | 35 | 8.03 | Dr. S. Flögel |
| LR1 – LR6 | Lopphavet, Norwegian Sea | 70°26.59'N 21°10.00'E | 230 | 6.5 | 35.1 | 8.05 | Raddatz et al. (2013) |
| LW1 – LW5 | Lopphavet, Norwegian Sea | 70°26.59'N 21°10.00'E | 230 | 6.5 | 35.1 | 8.05 | Raddatz et al. (2013) |
| NL-1 – NL6 | Nordleksa, Norwegian Sea | 63°36.43'N 9°22.66'E | 155 | 7.8 | 35 | 8.03 | Dr. S. Flögel |
| OS1 – OS6 | Oslo Fjord, North Sea | 59°04.01'N 10°44.31'E | 115 | 8.2 | 35.2 | 7.98 | Raddatz et al. (2013) |
| P391-1 – P391-3 | Lopphavet, Norwegian Sea | 70°26.59'N 21°10.00'E | 230 | 6.5 | 35.1 | 8.05 | Raddatz et al. (2013) |
| RL1-1 – RL1-4 | Trondheimfjord, Norwegian Sea | 63°28.61'N 9°59.72'E | 240 | 8.1 | 31.2 | 8.03 | Dullo et al. (2008), Rüggeberg et al. (2011) |
| RL4-1 – RL4-4 | Trondheimfjord, Norwegian Sea | 63°28.61'N 9°59.72'E | 240 | 8.1 | 31.2 | 8.03 | Dullo et al. (2008), Rüggeberg et al. (2011) |
| RL9-1 – RL9-5 | Trondheimfjord, Norwegian Sea | 63°28.61'N 9°59.72'E | 240 | 8.1 | 31.2 | 8.03 | Dullo et al. (2008), Rüggeberg et al. (2011) |
| SJ1 – SJ6 | Stjernsund Fjord, Norwegian Sea | 70°16.04'N 22°27.37'E | 295 | 5.9 | 35 | 8.20 | Rüggeberg et al. (2011) |
| SJ2-1 – SJ2-4 | Stjernsund Fjord, Norwegian Sea | 70°16.04'N 22°27.37'E | 295 | 5.9 | 35 | 8.20 | Rüggeberg et al. (2011) |

| | | | | | | | |
|---|---|---|---|---|---|---|---|
| TF1 – TF2 | Tautra Reef, Norwegian Sea | 63°35.36'N 10°31.23'E | 39 | 6 | 30.1 | 7.99 | Neulinger (2008) |
| TF2-1 – TF2-6 | Trondheimfjord, Norwegian Sea | 63°28.61'N 9°59.72'E | 240 | 8.1 | 31.2 | 7.99 | Dullo et al. (2008), Rüggeberg et al. (2011) |
| EFC1 – EFC6 | Sula Reef, Norwegian Sea | 64°05.93'N 8°05.47'E | 286 | 7.6 | 35.3 | 8.03 | Raddatz et al. (2013) |
| EFE1 – EFE4 | Sula Reef, Norwegian Sea | 64°05.93'N 8°05.47'E | 286 | 7.6 | 35.3 | 8.08 | Raddatz et al. (2013) |

**Table S1 Sample ident and location Metadata. Sample names such as A1-A4 relate to multiple samples that derive from the same coral specimen/calice.**

| Samplename | Species | Na/Ca [mmol/mol] | Mg/Ca [mmol/mol] | Sr/Ca [mmol/mol] |
|---|---|---|---|---|
| A1 - A4 | D | 29.02 | 3.83 | 10.23 |
| B1 - B2 | D | 24.87 | 3.17 | 9.91 |
| C1 - C2 | D | 29.25 | 3.97 | 9.95 |
| D1 – D4 | D | 27.57 | 3.94 | 10.01 |
| E1 - E3 | D | 24.14 | 4.03 | 9.84 |
| F1 – F4 | D | 24.61 | 3.56 | 10.05 |
| G1 – G4 | D | 31.04 | 4.25 | 10.18 |
| H1 – H3 | D | 23.93 | 3.62 | 10.21 |
| I1 – I3 | D | 23.75 | 3.99 | 9.65 |
| J1 – J2 | D | 21.2 | 2.45 | 10.35 |
| K1 – K5 | D | 28.72 | 7.84 | 9.6 |
| LO1 | D | 26.74 | 3.94 | 10 |
| MSS1 | D | 26.85 | 5.41 | 10.03 |
| MA1 | M | 25.18 | 3.73 | 10.47 |

| | | | | |
|---|---|---|---|---|
| MSM1 | D | 66.24 | 7.71 | 10.31 |
| MSM2 | D | 30.17 | 6.38 | 10.42 |
| MSM3 | M | 56.92 | 9.98 | 10.62 |
| MSM4 | D | 25.43 | 3.04 | 10.22 |
| ALB1 | D | 49.9 | 3.82 | 10.08 |
| KRS1 | C | 45.84 | 4.37 | 9.94 |
| KRS2 | C | 22.33 | 4.47 | 10.15 |
| KRS3 | C | 20.91 | 4.73 | 9.46 |
| KRS4 | C | 20.49 | 2.81 | 9.78 |
| KRS5 | C | 20.78 | 3.49 | 10.36 |
| L10-1 | D | 21.12 | 2.2 | 10 |
| L2-1 – L2-2 | D | 24.61 | 3.04 | 10.11 |

| | | | | |
|---|---|---|---|---|
| L3-1 – L3-2 | D | 24.58 | 3.19 | 10.36 |
| L5-1 – L5-2 | D | 23.43 | 3.1 | 10.39 |
| L6-1 – L6-2 | D | 23.79 | 3.68 | 10.26 |
| L7-1 – L7-2 | D | 25.45 | 3.49 | 10.41 |
| L8-1 – L8-2 | D | 23.67 | 3.23 | 10.21 |
| LR1 – LR6 | D | 27.18 | 3.34 | 10.31 |
| LW1 – LW5 | D | 26.23 | 3.79 | 9.63 |
| NL-1 – NL6 | D | 28.14 | 4.67 | 9.9 |
| OS1 – OS6 | D | 24.81 | 5.73 | 9.7 |
| P391-1 – P391-3 | D | 24.45 | 6.26 | 10.38 |
| RL1-1 – RL1-4 | D | 21.32 | 3.03 | 10.27 |
| RL4-1 – RL4-4 | D | 24.36 | 4.72 | 10.38 |
| RL9-1 – RL9-5 | D | 22.68 | 3.41 | 10.26 |
| SJ1 – SJ6 | D | 28.64 | 4.51 | 10.07 |
| SJ2-1 – SJ2-4 | D | 29.38 | 5.37 | 10.42 |

| | | | | |
|---|---|---|---|---|
| TF1 – TF2 | D | 25.72 | 6.1 | 10.13 |
| TF2-1 – TF2-6 | D | 21.42 | 3.78 | 10.2 |
| EFC1 – EFC6 | D | 26.01 | 4.31 | 9.94 |
| EFE1 – EFE4 | D | 27.1 | 3.8 | 10.31 |

**Table S2 Results of the ICP-OES measurements. Red values are identifed as outliers and not considered for further calculations. Species abreviations stand for D=*Desmophyllum pertusum* , M=*Madrepora oculata* and C=Caryophylliidae. Sample names such as A1-A4 relate to multiple samples that derive from the same coral specimen/calice.**